# Efficient RAW Image Deblurring with Adaptive Frequency Modulation

**Wenlong Jiao**[1]   **Binglong Li**[1]   **Wei Shang**[2]   **Ping Wang**[1]   **Dongwei Ren**[3]*
[1] School of Mathematics, Tianjin University
[2] School of Computer Science and Technology, Harbin Institute of Technology
[3] School of Artificial Intelligence, Tianjin University
{wenlong, li_binglong, wang_ping, rendw}@tju.edu.cn
{csweishang, rendongweihit}@gmail.com

## Abstract

Image deblurring plays a crucial role in enhancing visual clarity across various applications. Although most deep learning approaches primarily focus on sRGB images, which inherently lose critical information during the image signal processing pipeline, RAW images, being unprocessed and linear, possess superior restoration potential but remain underexplored. Deblurring RAW images presents unique challenges, particularly in handling frequency-dependent blur while maintaining computational efficiency. To address these issues, we propose Frequency Enhanced Network (FrENet), a framework specifically designed for RAW-to-RAW deblurring that operates directly in the frequency domain. We introduce a novel Adaptive Frequency Positional Modulation module, which dynamically adjusts frequency components according to their spectral positions, thereby enabling precise control over the deblurring process. Additionally, frequency domain skip connections are adopted to further preserve high-frequency details. Experimental results demonstrate that FrENet surpasses state-of-the-art deblurring methods in RAW image deblurring, achieving significantly better restoration quality while maintaining high efficiency in terms of reduced MACs. Furthermore, FrENet's adaptability enables it to be extended to sRGB images, where it delivers comparable or superior performance compared to methods specifically designed for sRGB data. The source code and pre-trained models are publicly available at https://github.com/WenlongJiao/FrENet.

## 1   Introduction

Image blur remains a pervasive challenge in computational photography, critically degrading visual quality and impeding downstream vision tasks. While deep learning has revolutionized image deblurring, most methods focus on processed sRGB images [27, 46, 49, 13, 22, 6], which suffer from irreversible information loss during the image signal processing (ISP) pipeline processing, including dynamic range compression and nonlinear transformations [1]. In contrast, RAW sensor data preserves linearity and high dynamic range, offering superior restoration potential through direct processing before ISP-induced degradations [42]. Despite this advantage, RAW image deblurring remains underexplored and faces challenges in effectively handling frequency-dependent blur patterns and maintaining computational efficiency.

Recent advancements in sRGB deblurring highlight the efficacy of CNNs [5] and Transformers [46] for spatial domain processing. However, directly applying these techniques to RAW data proves

---

*Corresponding author.

39th Conference on Neural Information Processing Systems (NeurIPS 2025).

suboptimal due to fundamental differences in noise characteristics and frequency response between the two domains [9]. Furthermore, spatial-domain methods may overlook the intrinsic relationship between blur formation and frequency domain representations, where convolutional degradations appear as multiplicative perturbations [18]. While some frequency-aware architectures have emerged for sRGB restoration [26], they often rely on computationally expensive attention mechanisms, unsuitable for RAW processing. They also lack adaptive spectral modulation capabilities that are necessary for capturing the complex frequency characteristics of RAW data.

To address these issues, we propose Frequency Enhanced Network (FrENet), a specialized RAW-to-RAW deblurring framework. Specifically, FrENet is built upon a U-Net architecture that integrates spatial and frequency domain processing to achieve efficient RAW image deblurring from three key perspectives. First, the core contribution of FrENet lies in a novel Adaptive Frequency Positional Modulation (AFPM) module, which dynamically adjusts frequency components according to their spectral positions. By leveraging a lightweight MLP to learn position-dependent modulation kernels, AFPM ensures precise control over critical frequency bands for detail recovery. Second, we incorporate frequency domain skip connections to preserve high-frequency details that are often lost during spatial downsampling. Third, our design prioritizes computational efficiency by adopting a compact CNN-based architecture, thereby avoiding computationally expensive Transformer operations while delivering superior performance.

Experiments demonstrate that our FrENet establishes a new state-of-the-art method for RAW image deblurring, achieving a 0.69dB PSNR improvement while requiring 75% fewer MACs compared to LoFormer-L [26] on the Deblur-RAW dataset [20]. Notably, FrENet exhibits remarkable adaptability. When applied to sRGB deblurring without any architectural modifications, it outperforms specialized sRGB deblurring methods on the RealBlur dataset [29] by 0.97dB PSNR (on RealBlur-J). These results validate our core contribution: adaptive frequency modulation enables both superior restoration quality and computational efficiency.

Our main contributions are three-fold:

- We propose a dedicated RAW-to-RAW deblurring network that systematically integrates spatial and frequency domain processing, explicitly leveraging the linearity and full spectral information of RAW data through learnable frequency modulation.

- A novel Adaptive Frequency Positional Modulation module is introduced to dynamically calibrate frequency components using location-dependent kernels via spectral position encoding.

- An efficiency-optimized network architecture is adopted, featuring frequency-aware skip connections for detail preservation across scales and compact convolution-Fourier hybridization to reduce computational costs.

## 2 Related Work

### 2.1 Image Deblurring in sRGB Domain

Deep learning methods have largely dominated the field of image deblurring by learning end-to-end mappings from blurred to sharp images, effectively tackling the challenging blind deblurring problem.

**CNN-based sRGB Deblurring:** Early deep learning approaches utilized Convolutional Neural Networks (CNNs) [38, 30, 32] often alongside traditional techniques. More recent CNN architectures moved towards purely end-to-end training, employing multi-scale strategies [27, 33, 7] or dynamic mechanisms [48, 12] to handle varying blur levels and scales. Networks like MPRNet [45] and HINet [4] further advanced performance through multi-stage refinement and sophisticated feature fusion. Notably, NAFNet [5] showcased the potential of a simple yet highly optimized U-shaped CNN architecture for efficient and effective deblurring by focusing on basic building blocks. Inspired by NAFNet's efficiency, our network utilizes a similar lightweight CNN backbone. However, most of these CNN-based methods primarily operate in the spatial domain and are designed and trained on sRGB data, facing a significant domain gap when applied directly to RAW data due to differences in noise, dynamic range, and processing pipeline.

**Transformer-based sRGB Deblurring:** Vision Transformers (ViTs) and their variants have been adapted for sRGB deblurring [3], leveraging their capability to model long-range dependencies

crucial for global blur patterns. To reduce the high computational cost of standard attention for high-resolution images, efficient Transformer designs like window-based attention [21], depth-wise convolution-based attention [46], and localized attention schemes [37, 35] have been proposed. Some recent Transformer-based methods, such as FFTformer [18] and Loformer [26], have explored incorporating frequency domain analysis within their architectures, demonstrating benefits for sRGB restoration. While these methods show promise in leveraging frequency information, they are tailored for the sRGB domain, are typically based on computationally different Transformer architectures, and may not offer the fine-grained, adaptive frequency modulation capability required to address the complex frequency characteristics and noise found in RAW data.

## 2.2 RAW Image Restoration

Operating directly on RAW sensor data offers significant advantages for image restoration tasks. Unlike sRGB data which has undergone irreversible processing by the camera's ISP, RAW data preserves linear intensity, high dynamic range, and richer original information, providing a better foundation for comprehensive restoration [1, 42].

**RAW Image Deblurring:** While traditional methods [34, 51] explored RAW image deblurring, deep learning research specifically for RAW-to-RAW deblurring is less developed than in the sRGB domain. Liang et al. [20] pioneered this deep learning sub-area with an end-to-end framework and dataset, processing both packed multi-channel and original single-channel RAW data. ELMformer [23] later introduced an efficient Transformer for RAW restoration, including deblurring on single-channel inputs. More recently, RawIR [10] addressed realistic RAW degradation synthesis and proposed a model for joint denoising and deblurring. While these deep learning works confirm the advantages of RAW domain deblurring, they primarily operate spatially. Crucially, they often lack fine-grained, adaptive processing of frequency components—vital for effectively separating blur and noise from scene details in the RAW data's frequency domain. Other works [11, 50, 28] explore RAW data for related tasks like joint processing pipelines but do not focus on general RAW-to-RAW deblurring with explicit frequency analysis.

**Other RAW Restoration Tasks:** The potential of processing RAW data has been successfully demonstrated in other image restoration tasks, including denoising [2, 14, 17, 44], super-resolution [39, 40, 19, 43, 16], and low-light enhancement [2, 15, 47, 41]. These studies collectively underscore the superior restorability offered by the RAW format. However, they address different types of degradations (primarily noise or resolution) and largely rely on spatial domain processing techniques. Our work specifically targets the deblurring problem and explores the relatively unutilized potential of integrating adaptive frequency analysis within the RAW domain for this task.

In summary, while deep learning has achieved significant success in sRGB deblurring, including recent explorations[18, 26, 25] into frequency domain processing within Transformer architectures[36], deep learning research for RAW image deblurring is less mature. Furthermore, existing RAW deblurring methods have not fully exploited the benefits of adaptive frequency domain analysis for better separation and restoration of details from blur and noise. Our proposed FrENet aims to fill this gap by presenting a novel RAW-to-RAW deblurring framework that combines an efficient CNN backbone with a unique adaptive frequency perception module designed to leverage the specific challenges and characteristics of RAW data in the frequency domain.

## 3 Proposed Method

### 3.1 Overall Framework

Given that the frequency domain of images provides reliable information about blur patterns [24], we use the Fast Fourier Transform (FFT) for frequency-based image component analysis. Notably, image blurring, a convolution in the spatial domain, corresponds to element-wise multiplication in the frequency domain, which fundamentally simplifies the restoration problem in this domain. This allows us to retain high-frequency information in the original image and analyze blur patterns to guide image deblurring. Our method, Frequency Enhanced Net (FrENet), focuses on effective frequency-domain processing within the U-shape architecture, as shown in Fig. 1(a).

Our FrENet comprises $L$ scales, i.e., it contains $L$ encoder layers, $L$ decoder layers, and a bottleneck layer. It starts with an initial convolutional layer that converts the input blurry image $\boldsymbol{y} \in \mathbb{R}^{C_{in} \times H \times W}$

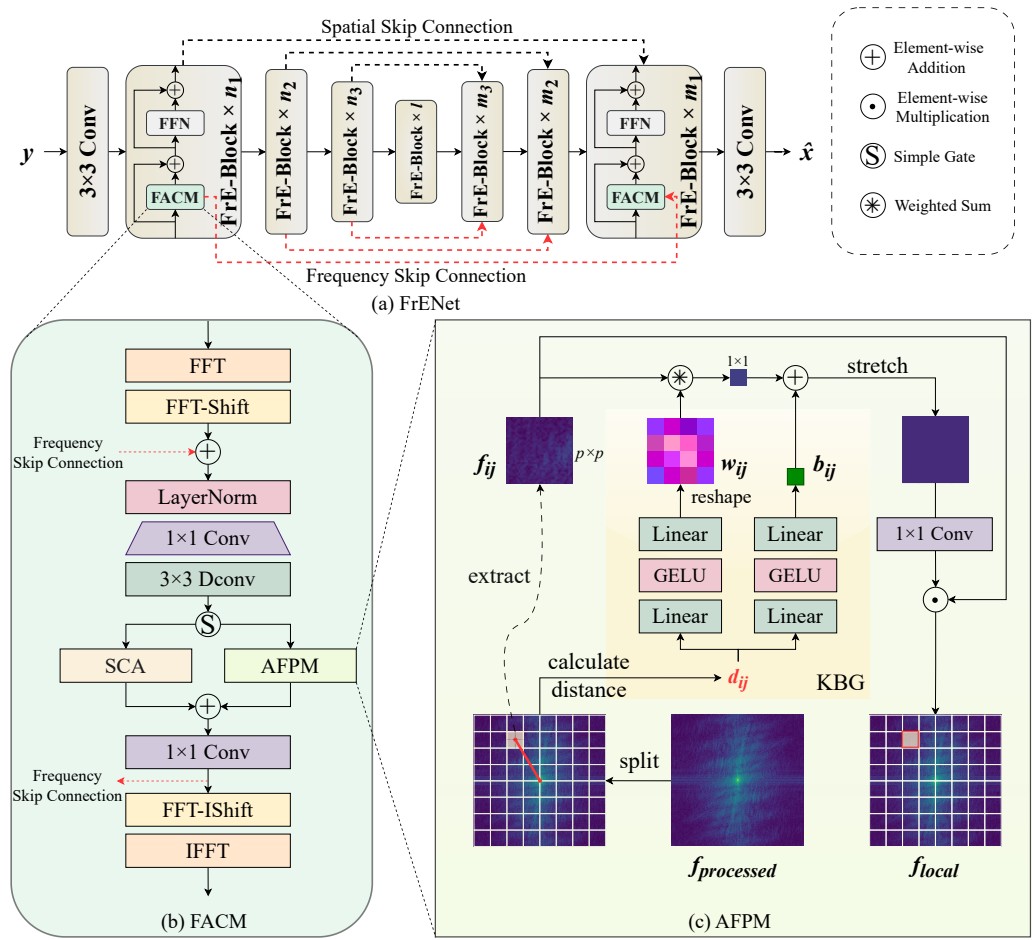

Figure 1: The architecture of our FrENet is designed with a compact convolution-Fourier hybrid structure, ensuring efficient inference. The key module FACM integrates global frequency information via SCA and local frequency details through AFPM. Notably, AFPM incorporates learnable frequency kernels determined by their spectral positions. Cooperated with frequency skip connections, AFPM substantially enhances frequency details, leading to significant performance improvements over state-of-the-art methods.

($C_{in} = 4$ for RAW image after packing) into higher-dimensional image features $\boldsymbol{f}_0^{enc} \in \mathbb{R}^{C \times H \times W}$ for subsequent processing. Each layer of the encoder, decoder, and bottleneck consists of multiple Frequency Enhanced Blocks (FrE-Blocks). The network ends with a final convolutional layer that converts image features of the last decoder layer $\boldsymbol{f}_0^{dec} \in \mathbb{R}^{C \times H \times W}$ into the final deblurred image $\hat{\boldsymbol{x}} \in \mathbb{R}^{C_{in} \times H \times W}$. The overall pipeline of our FrENet can be described as follows:

$$
\begin{aligned}
\boldsymbol{f}_0^{enc} &= \mathtt{Conv}_{3\times3}(\boldsymbol{y}), \\
\boldsymbol{f}_i^{enc} &= \mathtt{Encoder}_i(\boldsymbol{f}_{i-1}^{enc}) = \mathtt{FrE\text{-}Block}_{\times n_i}(\boldsymbol{f}_{i-1}^{enc}), i \in \{1, 2, \ldots, L\} \\
\boldsymbol{f}_L^{dec} &= \mathtt{Bottleneck}(\boldsymbol{f}_L^{enc}) = \mathtt{FrE\text{-}Block}_{\times l}(\boldsymbol{f}_L^{enc}), \\
\boldsymbol{f}_{i-1}^{dec} &= \mathtt{Decoder}_i(\boldsymbol{f}_i^{dec}) = \mathtt{FrE\text{-}Block}_{\times m_i}(\boldsymbol{f}_i^{dec}, \boldsymbol{f}_i^{enc}, \boldsymbol{f}_i'^{enc}), i \in \{L, L-1, \ldots, 1\} \\
\hat{\boldsymbol{x}} &= \mathtt{Conv}_{3\times3}(\boldsymbol{f}_0^{dec}),
\end{aligned}
\tag{1}
$$

where $n_i$ and $m_i$ represent the number of blocks in the $i$-th scale of the encoder layer and decoder layer, $l$ denotes the number of blocks in the bottleneck of the architecture. In the encoder, as the scale $i$ increases, the spatial resolution of the feature halves, and the channel number doubles, i.e., $\boldsymbol{f}_i^{enc} \in \mathbb{R}^{2^i C \times \frac{H}{2^i} \times \frac{W}{2^i}}$. In the decoder, this process is symmetric, with the spatial resolution of the feature doubling and channel number halving, i.e., $\boldsymbol{f}_i^{dec} \in \mathbb{R}^{2^i C \times \frac{H}{2^i} \times \frac{W}{2^i}}$. Our key innovations are:

1) Both frequency-domain and spatial-domain skip connections are set up between each encoder and decoder layer to feed the spatial feature $f_i^{enc}$ and frequency feature $f_i'^{enc}$ to the decoder layer. The frequency skip connection feature $f_i'^{enc}$ in the $i$-th scale is sourced from the $n_i$-th FrE-Block. This enables richer transmission of multi-scale spectral information along the encoder-decoder path. 2) Designing FrE-Blocks as the core processing units within each U-Net layer to directly handle frequency features, where both global and local frequency information can be well exploited for improving deblurring performance. We describe the details of the core unit FrE-Block in the following.

## 3.2 Frequency Enhanced Block

Each FrE-Block consists of a Frequency Adaptive Context Module (FACM) and a Feed-Forward Network (FFN) as shown in Fig. 1(a). For the structure of FFN, we adopt an implementation consistent with that used in Restormer [46]. The details of the FFN structure are provided in Appendix A.

The details of FACM are shown in Fig. 1(b). We take the first scale as an example. For simplicity, the input and output of each block are denoted as $f_{in}$ and $f_{out}$, respectively, without using subscripts. Given the feature in spatial domain $f_{in} \in \mathbb{R}^{C \times H \times W}$, we first adopt the FFT operation to convert the image from the spatial domain to the frequency domain and use the shift function FFT-Shift to recenter the zero-frequency component of the spectrum to the middle. If the FrE-Block is in a decoder layer, we adopt frequency skip-connections. The initial FFT-transformed feature is summed with the frequency domain feature from the corresponding encoder layer's last FrE-Block (which is the frequency domain output before IFFT), and the result is used as $f_{freq}$. If the FrE-Block is in an encoder layer or the bottleneck layer, the FFT-transformed features are directly used as $f_{freq}$. Considering that the values of features become complex numbers after the FFT, we concatenate their real and imaginary parts along the channel dimension. To mitigate the significant data distribution changes inherent in FFT, layer normalization (LN) is applied to stabilize the frequency features:

$$f_{norm} = \text{LN}(\text{Concat}(\mathfrak{R}(f_{freq}), \mathfrak{I}(f_{freq}))) \tag{2}$$

where $\mathfrak{R}, \mathfrak{I}$ represents the real and imaginary parts, respectively.

Then, the frequency features $f_{norm}$ undergo initial shared processing designed to extract foundational frequency patterns. A $1 \times 1$ convolution is first applied for channel-wise information fusion, such as between real and imaginary components. Next, a $3 \times 3$ depthwise convolution is applied to calculate local correlations between frequency bands in the frequency domain. A SimpleGate (SG) activation function [5] is then applied to obtain intermediate feature $f_{processed}$:

$$f_{processed} = \text{SG}(\text{DConv}_{3 \times 3}(\text{Conv}_{1 \times 1}(f_{norm}))) \tag{3}$$

The resulting feature map $f_{processed}$ is then processed through two parallel branches to capture both local details and global context within the frequency domain: 1) Local Frequency Feature Enhancement Branch and 2) Global Frequency Context Branch.

**Local Frequency Feature Enhancement Branch:** This branch feeds $f_{processed}$ into our proposed AFPM module. Due to the fact that different positions in the frequency domain represent image signals of different frequency bands, the proposed AFPM adaptively modulates features by generating position-aware weights. It adjusts features across different frequency bands based on their locations in the frequency domain, unlike traditional approaches that often apply uniform operations in different positions. To achieve this, AFPM employs position-sensitive, learnable operations that enhance the representation of specific frequency bands. We first divide the frequency feature map $f_{processed} \in \mathbb{R}^{C \times H \times W}$ into multiple feature patches, each sized $C \times p \times p$:

$$f_{processed} = \begin{bmatrix} f_{11} & \cdots & f_{1n} \\ \vdots & \ddots & \vdots \\ f_{m1} & \cdots & f_{mn} \end{bmatrix}, \quad m = \frac{H}{p}, n = \frac{W}{p} \tag{4}$$

We use the center of each patch as its position index $ij$ to indicate its relative location in the original feature map. The distance from the patch to the center of the feature map is defined as $d_{ij}$ as shown in Fig. 1(c). We apply two Kernel-Bias Generators (KBGs) to dynamically generate two position-specific components, i.e., position-aware kernels and biases conditioned on distance $d_{ij} \in \mathbb{R}^{1 \times 1 \times 1}$.

KBG contains two fully connected layers and GELU activations in between. By feeding the distance $d_{ij}$ into two KBGs, we can obtain position-aware kernels $w_{ij} \in \mathbb{R}^{1 \times p \times p}$ and biases $b_{ij} \in \mathbb{R}^{1 \times 1 \times 1}$, respectively. The kernels and biases adaptively adjust the significance of local frequency features for the current restoration task. Each kernel and bias is shared across channel dimension. Subsequently, following this kernel-bias application, we apply a $1 \times 1$ convolution across the channel dimension to further refine and adjust the channel features. Taking the patch with index $ij$ as an example, we use a weighted sum operation to combine the kernel with the frequency-domain patch and add the bias to adaptively adjust the frequency-domain features. This process is expressed as follows:

$$\text{AFPM}(\boldsymbol{f}_{ij}) = (\text{Conv}_{1 \times 1}(\boldsymbol{w}_{ij} * \boldsymbol{f}_{ij} + \boldsymbol{b}_{ij})) \odot \boldsymbol{f}_{ij} \tag{5}$$

where $*$ represents the weighted sum operation, $\odot$ represents the element-wise multiplication. Notably, we can compute all patches in parallel and place them back according to their index to obtain $\boldsymbol{f}_{local}$. More information of AFPM is provided in Appendix B.

**Global Frequency Context Branch:** Operating in parallel, this branch takes the same features as input, and applies Simplified Channel Attention (SCA) mechanism from NAFNet [5]. It aggregates global spatial information across the frequency map and computes channel-wise attention weights, allowing the network to recalibrate features based on global frequency context. It can be represented as:

$$\boldsymbol{f}_{global} = \text{SCA}(\boldsymbol{f}_{processed}) = (\text{Conv}_{1 \times 1}(\text{AvgPool}(\boldsymbol{f}_{processed}))) \odot \boldsymbol{f}_{processed} \tag{6}$$

where AvgPool represents the average pooling.

The features from the local branch $\boldsymbol{f}_{local}$ and the global branch $\boldsymbol{f}_{global}$ are then fused, typically via element-wise addition:

$$\boldsymbol{f}_{fused} = \boldsymbol{f}_{local} + \boldsymbol{f}_{global} \tag{7}$$

This fused feature map $\boldsymbol{f}_{fused}$ is passed through a final $1 \times 1$ convolution for further feature integration and potential dimensionality adjustment. Lastly, the processed frequency-domain features are first shifted by the inverse shift function FFT-IShift to reverse the centering of the zero-frequency component in the frequency domain, ensuring the reconstructed spatial-domain image is correct. If this FrE-Block is the last block of the current encoder layer, the complex-valued output at this stage is stored for the frequency domain skip connection to the corresponding decoder layer. Before applying IFFT, since the preceding operations are performed on the concatenated real and imaginary parts of the frequency-domain features, the processed feature map is first split into two equal halves along the channel dimension. These two halves are then interpreted as the real and imaginary parts, respectively, and are combined to form a complex-valued tensor, which is then transformed back to the spatial domain using IFFT, yielding the output feature map $\boldsymbol{f}_{out}$.

## 4 Experiments

### 4.1 Experiment Setup

**Model Configuration:** We evaluated two model configurations based on the FrENet architecture: **FrENet** was configured with a feature width of 32 and 24 processing FrE-Blocks, and **FrENet+** employed a feature width of 64 and 20 FrE-Blocks. Within every AFPM, we divide the feature map into an $8 \times 8$ grid of non-overlapping patches. If downsampling leads to very small feature map sizes ($< 8 \times 8$), we adopt a coarser granularity in this layer.

**Dataset:** We evaluate our method on five datasets: Deblur-RAW [20] in the RAW domain, and GoPro [27], HIDE [31], RealBlur-R, and RealBlur-J [29] in the sRGB domain. For the HIDE dataset, we specifically evaluate our method using the model trained on the GoPro dataset.

**Implementation Details:** For the Deblur-RAW [20] dataset, we adopted the preprocessing methodology used by RawNet [20]. This involved subtracting the black level and subsequently normalizing the raw data to the range [0, 1] by dividing by the maximum signal value. During training, $128 \times 128$ patches were randomly cropped from the normalized single-channel raw images. These single-channel patches, containing the RGGB Bayer pattern, were then packed into a 4-channel format which served as the network input. We employed the Adam optimizer with a batch size of 16. The initial learning rate was set to 0.001 and decayed using a cosine annealing scheduler over 1000 training epochs. **All models evaluated on the Deblur-RAW dataset were trained by us on NVIDIA RTX 5880 Ada Generation GPU.**

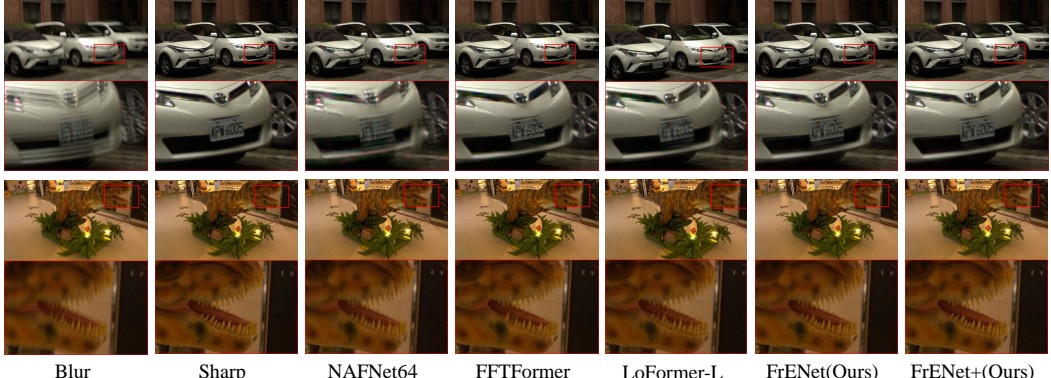

| Blur | Sharp | NAFNet64 | FFTFormer | LoFormer-L | FrENet(Ours) | FrENet+(Ours) |

Figure 2: Visual results of RAW blurry images. The images are visualized after being processed into sRGB domain using the LibRaw pipeline.

For extension to sRGB images, we only changed $C_{in} = 3$ in FrENet without any other modifications. For the RealBlur [29] dataset, we used the same settings as Deblur-RAW except that training was conducted using $256 \times 256 \times 3$ patches. For the GoPro dataset [27], we adopted training configurations from NAFNet [5].

For evaluation on test sets, we employed the sliding window strategy [8] to process full-resolution images. The sliding window size was equal to the training patch size, and the overlap size was half of the window size. Specifically, for the RealBlur test set, we utilized the official image alignment method provided by the dataset creators [29], ensuring a fair comparison.

**Loss Function:** In terms of the loss function, we used a weighted sum of $\mathcal{L}_1$ loss and Frequency Reconstruction (FR) loss $\mathcal{L}_{fr}$: $\mathcal{L} = \mathcal{L}_1 + 0.01\mathcal{L}_{fr}$, where $\mathcal{L}_{fr} = ||\mathcal{F}(\hat{I}) - \mathcal{F}(I)||$, and $\hat{I}, I, \mathcal{F}$ represent the deblurred image, the ground-truth and FFT operator, respectively.

**Evaluation Metric:** We evaluate the performance of image restoration methods using the Peak Signal-to-Noise Ratio (PSNR) and Structural Similarity Index (SSIM) as primary image quality metrics. Model efficiency, including Multiply-Accumulate Operations (MACs) and the number of parameters (Params), is calculated using relevant Python libraries.

### 4.2 Main Results

#### 4.2.1 Evaluation on RAW Image Deblurring

Table 1: Comparison of different image restoration methods on the Deblur-RAW dataset. PSNR and SSIM values are average scores evaluated in the RAW domain. MACs and Params are calculated on $128 \times 128 \times 1$ patches.

| Methods | PSNR ↑ | SSIM ↑ | MACs(G) ↓ | Params(M) ↓ |
|---------|--------|--------|-----------|-------------|
| NAFNet64 [5] | 40.35 | 0.982 | 3.96 | 67.79 |
| DeepRFT [24] | 42.40 | 0.988 | 5.05 | **9.55** |
| Restormer [46] | 42.86 | 0.989 | 8.82 | 26.10 |
| Stripformer [35] | 42.97 | 0.991 | 10.62 | 19.71 |
| FFTFormer [18] | 43.96 | 0.991 | 8.22 | 14.88 |
| LoFormer-S [26] | 42.97 | 0.990 | 3.27 | 16.36 |
| LoFormer-L [26] | 44.04 | 0.992 | 8.98 | 48.98 |
| **FrENet(Ours)** | 44.73 | 0.993 | **2.22** | 19.76 |
| **FrENet+(Ours)** | **45.63** | **0.994** | 7.30 | 48.38 |

As shown in Table 1, our proposed methods, FrENet and FrENet+, demonstrate superior performance on the Deblur-RAW dataset. Our FrENet+ model achieves state-of-the-art image restoration quality with the highest PSNR (45.63 dB) and SSIM (0.994). Our FrENet model also delivers strong perfor-

mance (PSNR 44.73 dB, SSIM 0.993), surpassing prior methods like LoFormer-L and FFTFormer, while being remarkably efficient. FrENet achieves the lowest MACs (2.22 G) among all compared methods, presenting an excellent balance of quality and computation. FrENet has 19.76M parameters; FrENet+ offers higher quality with 48.38M parameters and 7.30G MACs.

### 4.2.2 The Advantage of RAW Image Deblurring

Table 2: Effectiveness of Deblurring in RAW vs. sRGB Domain. We compare two pipelines: (i) deblurring in the RAW domain before ISP conversion, and (ii) deblurring in the sRGB domain after ISP conversion. Both pipelines use the identical LibRaw ISP. PSNR/SSIM are evaluated in the final sRGB domain.

| Methods | Pipeline | PSNR ↑ | SSIM ↑ |
|---|---|---|---|
| LoFormer-L [26] | (i) | 35.66 | 0.9656 |
| LoFormer-L [26] | (ii) | 31.26 | 0.9565 |
| FrENet | (i) | 36.12 | 0.9683 |
| FrENet | (ii) | 31.39 | 0.9574 |

We compared RAW vs. sRGB deblurring. To ensure a fair comparison, we used an EXIF tool to copy the original metadata to the deblurred RAW and utilized an identical LibRaw ISP for both pipelines. Table 2 confirms the pre-ISP approach is superior, with FrENet gaining 4.73 dB PSNR. This shows that the ISP's lossy operations irretrievably degrade information, making pre-ISP restoration essential.

### 4.2.3 Evaluation on sRGB Image Deblurring

Table 3: Quantitative comparison on sRGB datasets. $^\star$ indicates that the results are inferred using the checkpoint provided by the author, which are not reported in [26].

| | Synthetic | | | | | Real-world | | |
|---|---|---|---|---|---|---|---|---|
| | **GoPro** | | **HIDE** | | **RealBlur-R** | | **RealBlur-J** | |
| **Methods** | PSNR ↑ | SSIM ↑ | PSNR ↑ | SSIM ↑ | PSNR ↑ | SSIM ↑ | PSNR ↑ | SSIM ↑ |
| Restormer [46] | 32.92 | 0.961 | 31.22 | 0.942 | 36.19 | 0.957 | 28.96 | 0.879 |
| DeepRFT+ [24] | 33.52 | 0.965 | 31.66 | 0.946 | 40.01 | 0.973 | 32.63 | 0.933 |
| FFTFormer [18] | **34.21** | 0.969 | 31.62 | 0.945 | 40.11 | 0.973 | 32.62 | 0.932 |
| LoFormer-B [26] | 33.99 | 0.968 | 31.71 | 0.948 | 40.23 | 0.974 | 32.90 | 0.933 |
| LoFormer-L [26] | 34.09 | 0.969 | 31.86 | 0.949 | 40.60$^\star$ | **0.976$^\star$** | 32.88$^\star$ | 0.936$^\star$ |
| **FrENet+(Ours)** | 34.11 | **0.969** | **31.92** | **0.949** | **40.74** | 0.975 | **33.87** | **0.939** |

Table 3 provides a quantitative comparison of FrENet+ on synthetic (GoPro, HIDE) and real-world (RealBlur-R, RealBlur-J) sRGB deblurring datasets. FrENet+ demonstrates strong performance across all benchmarks. On synthetic datasets, it achieves leading results on HIDE (PSNR 31.92 dB, SSIM 0.949) and competitive performance on GoPro (PSNR 34.11 dB, SSIM 0.969). Notably, our method excels on the challenging real-world datasets. FrENet+ sets a new state-of-the-art on RealBlur-R with the highest PSNR (40.74 dB) and SSIM (0.975), significantly outperforming prior methods. Similarly, on RealBlur-J, it achieves the highest PSNR (33.87 dB) and SSIM (0.939), showing a clear advantage. These results highlight FrENet+'s effectiveness and robustness for diverse sRGB image restoration tasks, particularly in real-world scenarios.

### 4.3 Ablation Study

We conduct a comprehensive ablation study to investigate the contribution of different components and design choices within our proposed FrENet, evaluated on the Deblur-RAW dataset. The results are presented in Table 5 and Table 6. Our study focuses on: (1) the necessity of frequency domain modeling, (2) the roles of Spatial and Frequency Skip Connections in FrENet, and (3) the effectiveness and internal mechanisms of FrE-Block.

**Necessity of Frequency Domain Processing:** To validate our core design choice of operating primarily in the frequency domain, we created a spatial-only analogue, FrENetSpatial. This was

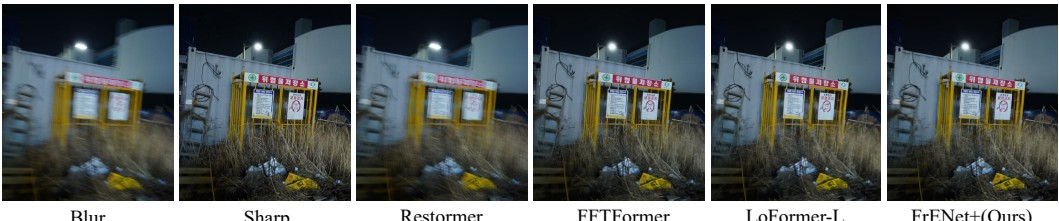

| Blur | Sharp | Restormer | FFTFormer | LoFormer-L | FrENet+(Ours) |

Figure 3: Visual results of real blurry images from RealBlur-J dataset.

Table 4: Ablation on the necessity of frequency domain processing.

| Method | PSNR | SSIM |
|---|---|---|
| FrENetSpatial | 41.89 | 0.9894 |
| FrENet (Ours) | **44.73** | **0.9931** |

achieved by removing all FFT and iFFT operations from our architecture, forcing all modules like convolutions to process features purely in the spatial domain. As shown in Table 4, the performance collapses: FrENetSpatial achieves only 41.89 dB PSNR, while our full FrENet reaches 44.73 dB. This massive 2.84 dB gain confirms that processing features within the frequency domain is the primary driver of our model's success.

Table 5: Ablation study of spatial and frequency skip connections.

| Spatial Skip Connection | Frequency Skip Connection | PSNR | SSIM |
|---|---|---|---|
| ✗ | ✓ | 43.91 | 0.9899 |
| ✓ | ✗ | 44.39 | 0.9927 |
| ✓ | ✓ | **44.73** | **0.9931** |

**Effectiveness of Frequency Domain Skip Connections:** We evaluate the contribution of the skip connections within the overall FrENet's UNet architecture. As shown in Table 5, the full architecture, incorporating both Spatial and Frequency skip connections, achieves 44.73 dB PSNR and 0.9931 SSIM. Ablating the proposed Frequency Skip Connection while retaining the standard Spatial Skip Connection leads to a noticeable performance decrease to 44.39 dB PSNR and 0.9927 SSIM. This demonstrates the significant positive impact of integrating frequency-domain information via the Frequency Skip Connection, complementing the spatial information from the traditional Spatial Skip Connection and highlighting a key architectural innovation of FrENet.

Table 6: Ablation study of FrE-Block.

| Method | Local Branch | Global Branch | Division Granularity | PSNR | SSIM |
|---|---|---|---|---|---|
| Average Pooling | ✓ | ✓ | $8 \times 8$ | 44.35 | 0.9927 |
| Adaptive | ✓ | ✓ | $2 \times 2$ | 44.44 | 0.9928 |
| | ✓ | ✓ | $4 \times 4$ | 44.52 | 0.9929 |
| | ✗ | ✓ | $8 \times 8$ | 44.48 | 0.9928 |
| | ✓ | ✗ | $8 \times 8$ | 44.67 | 0.9930 |
| | ✓ | ✓ | $8 \times 8$ | **44.73** | **0.9931** |

**Effectiveness of Local and Global Branches:** As shown in Table 6, we evaluate the interplay between our Local Branch and Global Branch under otherwise identical settings. Our full proposed model, combining both branches, achieves the best performance (44.73 dB PSNR, 0.9931 SSIM). Ablating the Global Branch and using only our Local Branch leads to a slight performance decrease (44.67 dB PSNR, 0.9930 SSIM), indicating the Global Branch's complementary contribution. Conversely, ablating our Local Branch and relying solely on the Global Branch results in a more

significant performance drop (44.48 dB PSNR, 0.9928 SSIM), highlighting the critical importance of our AFPM.

**Effectiveness of Division Granularity**: To investigate the influence of the division granularity within the FrE-Block's feature processing, we compared different division sizes while keeping both the Local and Global Branches active. As shown in Table 6, decreasing the granularity from $8 \times 8$ to $4 \times 4$ and further to $2 \times 2$ consistently leads to a performance drop. Specifically, changing from the optimal $8 \times 8$ division (64 patches) results in a decrease from 44.73 dB to 44.52 dB PSNR and 0.9931 to 0.9929 SSIM for the $4 \times 4$ division (16 patches), and even lower scores (44.44 dB PSNR, 0.9928 SSIM) for the $2 \times 2$ division (4 patches). This trend clearly indicates that a finer-grained division is more effective for extracting and modulating features, likely enabling the module to capture richer and more detailed local contextual and positional information within the feature map.

**Effectiveness of Adaptive Modulation:** As shown in Table 6, we compare our proposed adaptive modulation mechanism within the AFPM to a variant that employs a simpler, fixed pooling strategy for feature aggregation. The results indicate that the performance of the pooling-based approach is significantly inferior to that of our proposed adaptive modulation mechanism. This comparison not only validates the effectiveness of our AFPM but also highlights its superiority in adaptively processing frequency-domain features compared to the fixed pooling-based method.

## 5 Limitations

Despite achieving state-of-the-art deblurring performance and being more efficient than Transformer-based methods, FrENet has limitations. First, the heavy reliance on FFT and IFFT introduces substantial computational cost for high-resolution images. Second, the AFPM's simplified positional encoding may fail to fully capture complex spatial dependencies, indicating the need for more advanced techniques.

## 6 Conclusion

Based on the importance of analyzing image frequency characteristics for image restoration, we propose the Frequency-Enhanced U-Net (FrENet). FrENet integrates traditional spatial domain skip connections with frequency domain skip connections and operates directly on frequency features within its core processing unit, the FrE-Block. The FrE-Block contains a Frequency Adaptive Context Module that utilizes Adaptive Frequency Positional Modulation for local frequency details and Simplified Channel Attention for global spectral context, enabling precise modulation and utilization of frequency features to significantly improve image restoration performance. Furthermore, the adaptive frequency modulation mechanism demonstrates potential applicability to other image restoration tasks in RAW domain.

## Acknowledgements

This work was supported in part by the National Natural Science Foundation of China (62576241, 62172127, U23B2049 and U22B2035), the Natural Science Foundation of Heilongjiang Province (YQ2022F004).

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

# Appendices

## A   Details of FFN

Our FrE-Block utilizes a FFN module adapted from the Gated-Dconv FFN structure introduced in Restormer [46]. Given an input feature $\boldsymbol{f}_{in} \in \mathbb{R}^{C \times H \times W}$, the FFN processes it through two parallel branches followed by an element-wise product and a residual connection.

The first branch processes $\boldsymbol{f}_{in}$ through a $1 \times 1$ convolution for channel expansion, a $3 \times 3$ depth-wise convolution for spatial feature mixing within each channel group, and a GELU activation function. The second branch processes $\boldsymbol{f}_{in}$ through a $1 \times 1$ convolution followed by a $3 \times 3$ depth-wise convolution.

The outputs of these two branches are multiplied element-wise, acting as a content-aware gating mechanism that selectively modulates the features. Finally, a $1 \times 1$ convolution projects the features back to the original channel dimension, and a residual connection is added to the input $\boldsymbol{f}_{in}$.

The structure can be mathematically represented as:

$$\boldsymbol{f}_{out} = \boldsymbol{f}_{in} + \text{Conv}_{1\times1}(\text{GELU}(\text{DConv}_{3\times3}(\text{Conv}_{1\times1}(\boldsymbol{f}_{in})))) \odot (\text{DConv}_{3\times3}(\text{Conv}_{1\times1}(\boldsymbol{f}_{in}))) \quad (1)$$

where $\text{Conv}_{1\times1}$ denotes a $1 \times 1$ convolution (often with channel expansion/reduction internally), $\text{DConv}_{3\times3}$ is a $3 \times 3$ depth-wise convolution, GELU is the Gaussian Error Linear Unit activation, and $\odot$ represents element-wise multiplication. This structure allows the network to learn complex feature transformations while maintaining computational efficiency and capturing local spatial context.

# B Discussion of AFPM

This appendix provides a detailed discussion of the AFPM module, the core component of our FrE-Block designed for position-aware frequency feature refinement.

As discussed in the main text, different locations in the frequency domain represent image content at different spatial frequencies (low frequencies near the center, high frequencies towards the periphery). AFPM leverages this property to adaptively process features based on their spectral position.

The module takes a frequency feature map $\boldsymbol{f}_{processed} \in \mathbb{R}^{C \times H \times W}$ as input. We first divide $\boldsymbol{f}_{processed}$ into a grid of non-overlapping patches $\boldsymbol{f}_{ij} \in \mathbb{R}^{C \times p \times p}$, where $(i, j)$ indicates the patch's row and column index in the grid, and the grid has $m = H/p$ rows and $n = W/p$ columns.

For each patch $(i, j)$, we first compute its normalized Euclidean distance $\boldsymbol{d}_{ij} \in \mathbb{R}^{1 \times 1 \times 1}$ from the patch's center to the center of the entire feature map. This scalar distance $\boldsymbol{d}_{ij}$ serves as a proxy for the dominant frequency range covered by this patch and is used as input to the Kernel-Bias Generators.

AFPM then employs two lightweight Kernel-Bias Generators (KBGs). Each KBG is an MLP consisting of two linear layers with an intermediate GELU activation, implementing a non-linear mapping. These KBGs dynamically generate position-aware parameters conditioned on $\boldsymbol{d}_{ij}$:

- A kernel $\boldsymbol{w}_{ij} \in \mathbb{R}^{1 \times p \times p}$: Generated by $\text{KBG}^{kernel}(\boldsymbol{d}_{ij})$. This kernel provides spatial weights specific to the patch's frequency location and is shared across all $C$ channels.
- A bias $\boldsymbol{b}_{ij} \in \mathbb{R}^{1 \times 1 \times 1}$: Generated by $\text{KBG}^{bias}(\boldsymbol{d}_{ij})$. This provides a location-specific bias, also shared across channels.

These generated parameters are used to modulate the patch features. The modulation is performed as follows for each patch $\boldsymbol{f}_{ij}$:

$$\text{AFPM}(\boldsymbol{f}_{ij}) = (\text{Conv}_{1 \times 1}(\boldsymbol{w}_{ij} * \boldsymbol{f}_{ij} + \boldsymbol{b}_{ij})) \odot \boldsymbol{f}_{ij} \tag{2}$$

Here, $\boldsymbol{w}_{ij} * \boldsymbol{f}_{ij}$ denotes a channel-wise weighted summation: for each channel, the $p \times p$ feature slice of $\boldsymbol{f}_{ij}$ is element-wise multiplied by the shared $1 \times p \times p$ kernel $\boldsymbol{w}_{ij}$, and the results are then summed spatially, yielding an intermediate feature of size $\mathbb{R}^{C \times 1 \times 1}$. The position-aware bias $\boldsymbol{b}_{ij}$ (broadcast to $\mathbb{R}^{C \times 1 \times 1}$) is added to this aggregated feature. Subsequently, a $1 \times 1$ convolution processes this $\mathbb{R}^{C \times 1 \times 1}$ tensor. This convolution facilitates channel-wise interactions and transforms the aggregated, position-biased information into a refined per-channel modulation factor of size $\mathbb{R}^{C \times 1 \times 1}$. Finally, this modulation factor is broadcast back to the patch dimensions $\mathbb{R}^{C \times p \times p}$ and element-wise multiplied ($\odot$) with the original patch features $\boldsymbol{f}_{ij}$. This process allows AFPM to learn position-dependent, channel-wise scaling factors that adaptively enhance or suppress frequency components within each patch based on its location in the spectrum.

The adaptive, position-aware modulation performed by AFPM is visually demonstrated in Figure B.1 (which includes feature maps and generated kernels $\boldsymbol{w}_{i,j}$ in the RAW domain) and Figure B.2 (feature maps in the sRGB domain). The KBGs' behavior, central to AFPM's adaptivity, is particularly evident from the visualized kernels $\boldsymbol{w}_{i,j}$ in Figure B.1. These $1 \times p \times p$ kernels are shown for different spectral patch locations across various network stages ($\text{Encoder}_1$, $\text{Bottleneck}$, $\text{Decoder}_1$). In these kernel visualizations, darker colors indicate higher learned weight values, and lighter colors represent lower values. The indices $i, j$ in $\boldsymbol{w}_{i,j}$ (e.g., $\boldsymbol{w}_{0,0}, \boldsymbol{w}_{1,1}, \boldsymbol{w}_{2,2}, \boldsymbol{w}_{3,3}$ as shown in the figure) correspond to patches at varying normalized Euclidean distances $\boldsymbol{d}_{ij}$ from the center of the frequency map. For instance, $\boldsymbol{w}_{0,0}$ represents the kernel for a patch in a peripheral, high-frequency region (e.g., top-left if following standard image indexing), while kernels with higher indices like $\boldsymbol{w}_{3,3}$ (assuming a $4 \times 4$ display of kernels) are progressively closer to the center, corresponding to lower-frequency regions.

Observing the visualized kernels $\boldsymbol{w}_{i,j}$ and feature maps in Figure B.1 and Figure B.2 reveals several key aspects:

- **Positional Specificity of Kernels and Modulation:** The structure and intensity patterns of the kernels $\boldsymbol{w}_{i,j}$ in Figure B.1 visibly change with their spectral position. For example, in $\text{Encoder}_1$ of Figure B.1, the peripheral kernels (e.g., $\boldsymbol{w}_{0,0}, \boldsymbol{w}_{1,1}$) appear to have more distinct, higher-intensity (darker) patterns compared to the more central kernels (e.g.,

$\boldsymbol{w}_{2,2}, \boldsymbol{w}_{3,3}$), which might be smoother or have generally lower intensity values. This suggests that AFPM learns to apply stronger or more structured modulation to high-frequency components (potentially to enhance details or suppress specific noise patterns) and a different, perhaps more subtle or uniform, modulation to low-frequency components (to adjust overall brightness or contrast in those bands). The resulting $\text{AFPM}(\boldsymbol{f}_{processed})$ maps reflect these position-specific modulations. This directly confirms that the KBGs learn to generate distinct spatial weighting strategies based on the input distance $\boldsymbol{d}_{ij}$.

- **Layer-wise Adaptation and Effect on Features:** The characteristics of the learned kernels and their impact on feature maps adapt across different network layers and domains:

  - In early encoder stages (e.g., $\text{Encoder}_1$ in both Figure B.1 and B.2), AFPM appears to perform initial spectral refinement. The kernels might be learning to selectively boost certain structural frequencies or perform a gentle equalization. The $\text{AFPM}(\boldsymbol{f}_{processed})$ maps show subtle but widespread adjustments compared to $\boldsymbol{f}_{processed}$.

  - In the bottleneck (middle row of both figures), features are highly compressed. The kernels $\boldsymbol{w}_{i,j}$ (in Figure B.1) appear to adapt to this abstract representation, and the resulting $\text{AFPM}(\boldsymbol{f}_{processed})$ shows more pronounced, localized changes, likely focusing on preserving or transforming features critical for the decoder.

  - In decoder stages (e.g., $\text{Decoder}_1$, bottom row of both figures), the modulation is critical for reconstruction and often more aggressive. In Figure B.1 (RAW), the kernels $\boldsymbol{w}_{i,j}$ for $\text{Decoder}_1$ might learn to strongly amplify frequencies corresponding to edges while suppressing others. This is reflected in $\text{AFPM}(\boldsymbol{f}_{processed})$ where structured details appear sharpened. Notably, in Figure B.2 (sRGB) for $\text{Decoder}_1$, specific patterns in $\boldsymbol{f}_{processed}$ that resemble blur artifacts (e.g., diffuse diagonal bands or "ghosting") are visibly suppressed or transformed in $\text{AFPM}(\boldsymbol{f}_{processed})$, leading to a cleaner $\boldsymbol{f}_{out}$. This targeted suppression in the sRGB domain highlights AFPM's ability to adapt its frequency modulation to combat different manifestations of blur.

- **Content-Independent Nature of Kernels:** It is crucial to reiterate that the kernels $\boldsymbol{w}_{ij}$ themselves are generated based only on the patch's spectral position $\boldsymbol{d}_{ij}$, not its content $\boldsymbol{f}_{ij}$. The visualization of $\boldsymbol{w}_{ij}$ in Figure B.1 thus purely reflects the learned spatial modulation strategy for a given frequency location. The actual content adaptation occurs when this position-specific kernel modulates the patch features $\boldsymbol{f}_{ij}$ as per Equation 2.

This layer-dependent and position-specific modulation capability, evidenced by both the overall feature map transformations (Figure B.1, Figure B.2) and the varying structures of the generated kernels (Figure B.1), arises directly from the KBGs. By allowing the network to learn how to weight and shift frequency components based on where they are in the spectrum, AFPM provides a powerful and flexible mechanism for adaptive frequency domain processing.

In the main paper, our ablation studies on division granularity (comparing $2 \times 2$, $4 \times 4$, and $8 \times 8$ grids for patch division within AFPM) demonstrated that a finer-grained division generally leads to better performance, with the $8 \times 8$ grid yielding the best results among those tested. This suggests that enabling AFPM to operate on more localized spectral regions allows for more precise modulation. For instance, we also experimented with fixing the kernel size of $\boldsymbol{w}_{ij}$ to $4 \times 4$ while increasing the number of patches (i.e., making the grid finer than $8 \times 8$ such that each patch is smaller than $4 \times 4$ is not possible, rather, if the feature map is $H \times W$, a $k \times k$ grid means $p = H/k, W/k$. If $p$ is fixed at 4, then a larger $H, W$ means more patches). Conceptually, if each patch becomes very small (e.g., if $p$ itself was reduced, like using $p = 4$ for the patch size which $\boldsymbol{w}_{ij}$ operates on, and having more such patches in a larger grid), this could offer even more precise control, and preliminary tests indeed showed further improvements. However, this significantly increases the number of KBG evaluations (if each patch gets its own kernel) or the complexity of managing these smaller patches, leading to higher parameter counts and computational costs. Therefore, the $8 \times 8$ grid represented a good balance of performance and efficiency for our reported results.

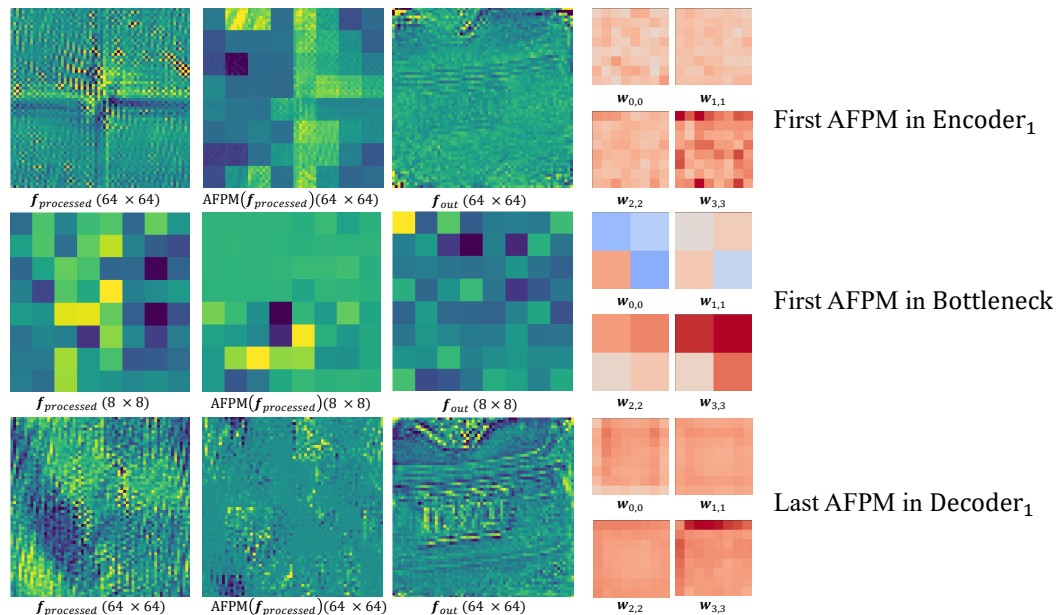

Figure B.1: Multi-level feature visualization of the key stages around AFPM in the RAW domain. Each row shows the input frequency feature map ( $f_{processed}$ ), the output after AFPM (AFPM($f_{processed}$)), the spatial domain output of the block ($f_{out}$) , and kernels $w_{i,j}$ generated by KBGs from a representative block in Encoder$_1$, Bottleneck, and Decoder$_1$. Deeper colors represent higher values.

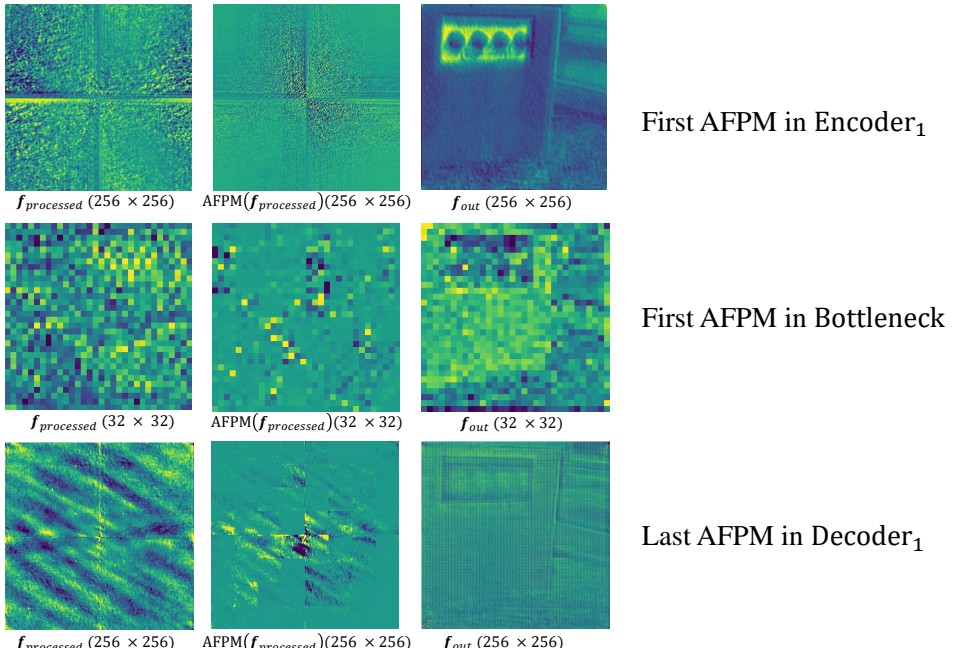

Figure B.2: Multi-level feature visualization of the key stages around AFPM in sRGB domain. Each row shows the input frequency feature map ( $f_{processed}$ ), the output after AFPM (AFPM($f_{processed}$)), and the spatial domain output of the block ($f_{out}$) from a representative block in Encoder$_1$, Bottleneck, and Decoder$_1$. Deeper colors represent higher values.

# C  Performance and Efficiency Analysis

## C.1  Inference Efficiency on Full-Resolution Images

While the main body of the paper analyzes computational efficiency on the $128\times128$ patches used during training, it is crucial to evaluate the model's practical performance on full-resolution images, which is the typical use case. To this end, we employ a standard sliding-window approach for inference. The high-resolution input image is partitioned into overlapping 128x128 patches, processed independently, and the results are then stitched together to reconstruct the final output. This strategy is a common practice in the field, also adopted by competing methods like LoFormer[26] and DeepRFT[24], ensuring a fair comparison.

We benchmarked the end-to-end inference time and GPU memory usage on full-resolution RAW images using a single NVIDIA RTX 5880 Ada GPU. As shown in Table C.1, the results confirm that our model's efficiency on small patches translates to a significant real-world advantage. FrENet is demonstrably faster ($1.36\times$ to $3\times$) and more memory-efficient than powerful Transformer-based baselines in this practical, end-to-end scenario.

Table C.1: Efficiency Comparison on Full-Resolution RAW Images.

| Methods | Params(M)↓ | GPU Memory(MB)↓ | Runtime(ms)↓ |
|---|---|---|---|
| Restormer [46] | 26.10 | 1238.71 | 102.95 |
| FFTFormer [18] | **14.88** | 2193.03 | 222.56 |
| LoFormer-L [26] | 48.98 | 2391.10 | 222.49 |
| **FrENet+(Ours)** | 19.76 | **1083.30** | **75.36** |

## C.2  Module-Level Efficiency Analysis

To provide a deeper understanding of the computational cost distribution within our model, we present a module-level analysis of a single FrE-Block. The analysis, detailed in Table C.2, reveals that our proposed core components, the AFPM and SCA modules, are highly parameter-efficient. Combined, they account for only 28.1% of the model's total parameters and a mere 1.8% of the MACs.

The majority of parameters and computational load are attributed to standard architectural components, such as the Feed-Forward Network (FFN) and convolutional layers. This breakdown underscores that our model's performance gains stem from the targeted and efficient design of its novel modules rather than an increase in overall model complexity.

Table C.2: Per-Module Cost Analysis of FrENet on $128 \times 128$ size patches.

| Module | Params(M) | MACs(G) | Runtime (ms) |
|---|---|---|---|
| Convolutional Layers | 6.0 (30.4%) | 0.92 (41.4%) | 0.149 (16.0%) |
| AFPM Module | 2.86 (14.4%) | 0.03 (1.4%) | 0.103 (11.1%) |
| SCA Module | 2.71 (13.7%) | 0.01 (0.4%) | 0.039 (4.2%) |
| Others (e.g., Layernorm) | 8.29 (41.9%) | 1.26 (56.7%) | 0.639 (68.7%) |
| Total | 19.76 | 2.22 | 0.93 |

## C.3    Visualization of Comparison

Figure C.3 presents a quantitative comparison of various image deblurring methods on the Deblur-RAW dataset, illustrating the trade-off between deblurring performance (PSNR, Y-axis) and computational efficiency (MACs, X-axis). The size of each bubble indicates the model's parameter count. Positioned in the upper-left region, our proposed method FrENet demonstrates superior performance at a significantly lower computational cost. Specifically, FrENet achieves state-of-the-art deblurring performance. Compared to competitive methods like LoFormer-L, which achieve comparable PSNR, FrENet requires considerably fewer MACs. This plot clearly showcases FrENet's effectiveness in balancing high restoration quality and computational efficiency on the Deblur-RAW dataset.

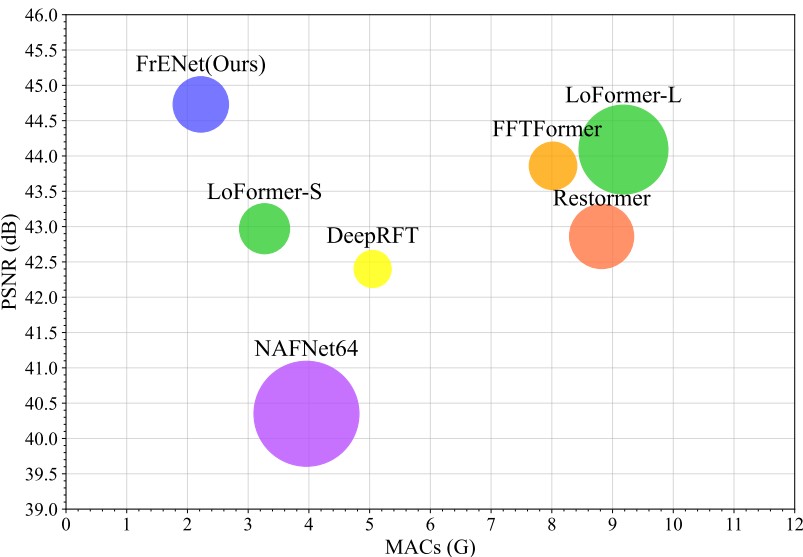

Figure C.3: Performance (PSNR) and efficiency (Params, MACs) comparison of various image deblurring methods on the Deblur-RAW dataset.

# D    Visual Results

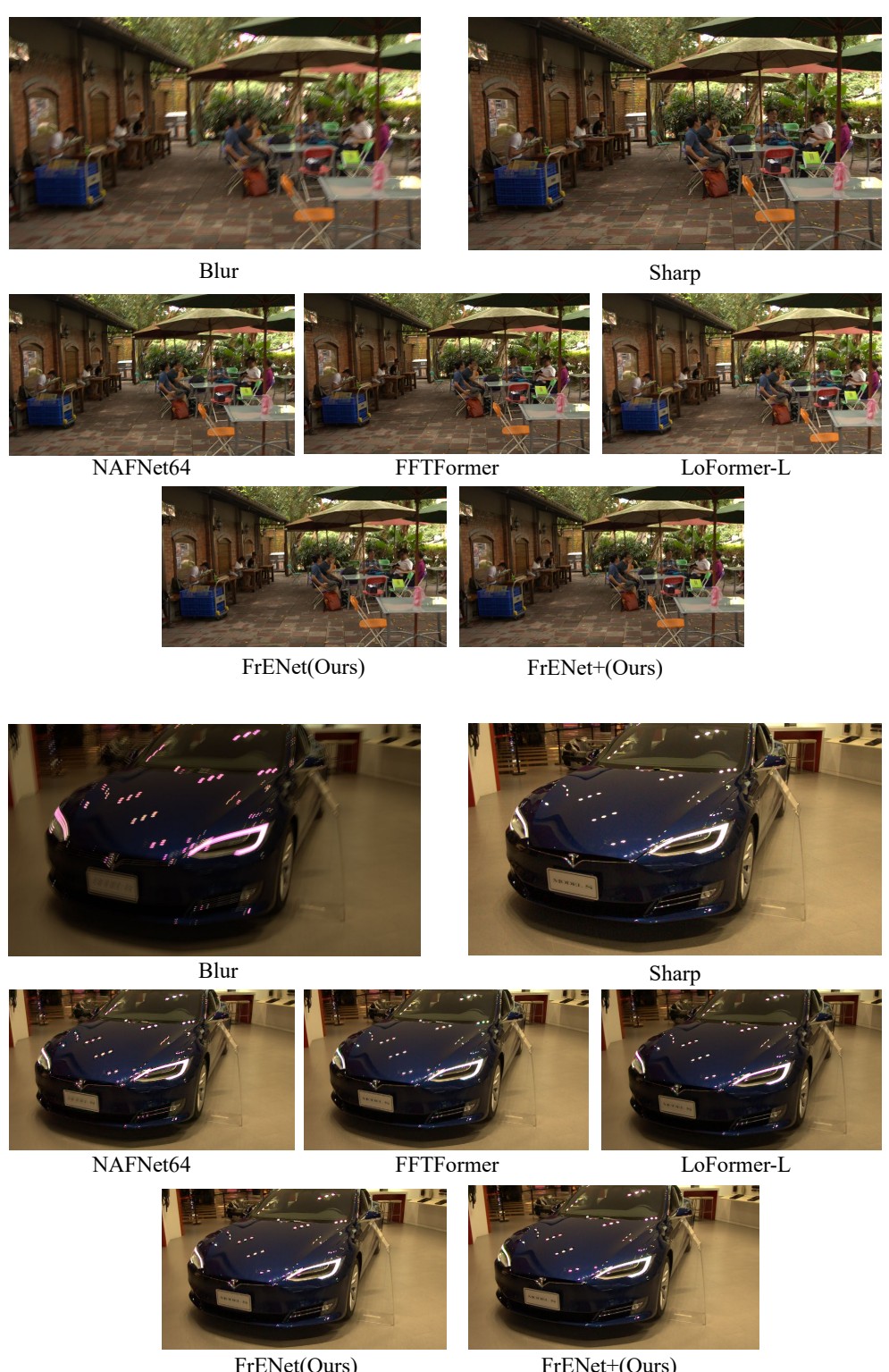

Figure D.4: Visual results on the Deblur-RAW dataset.

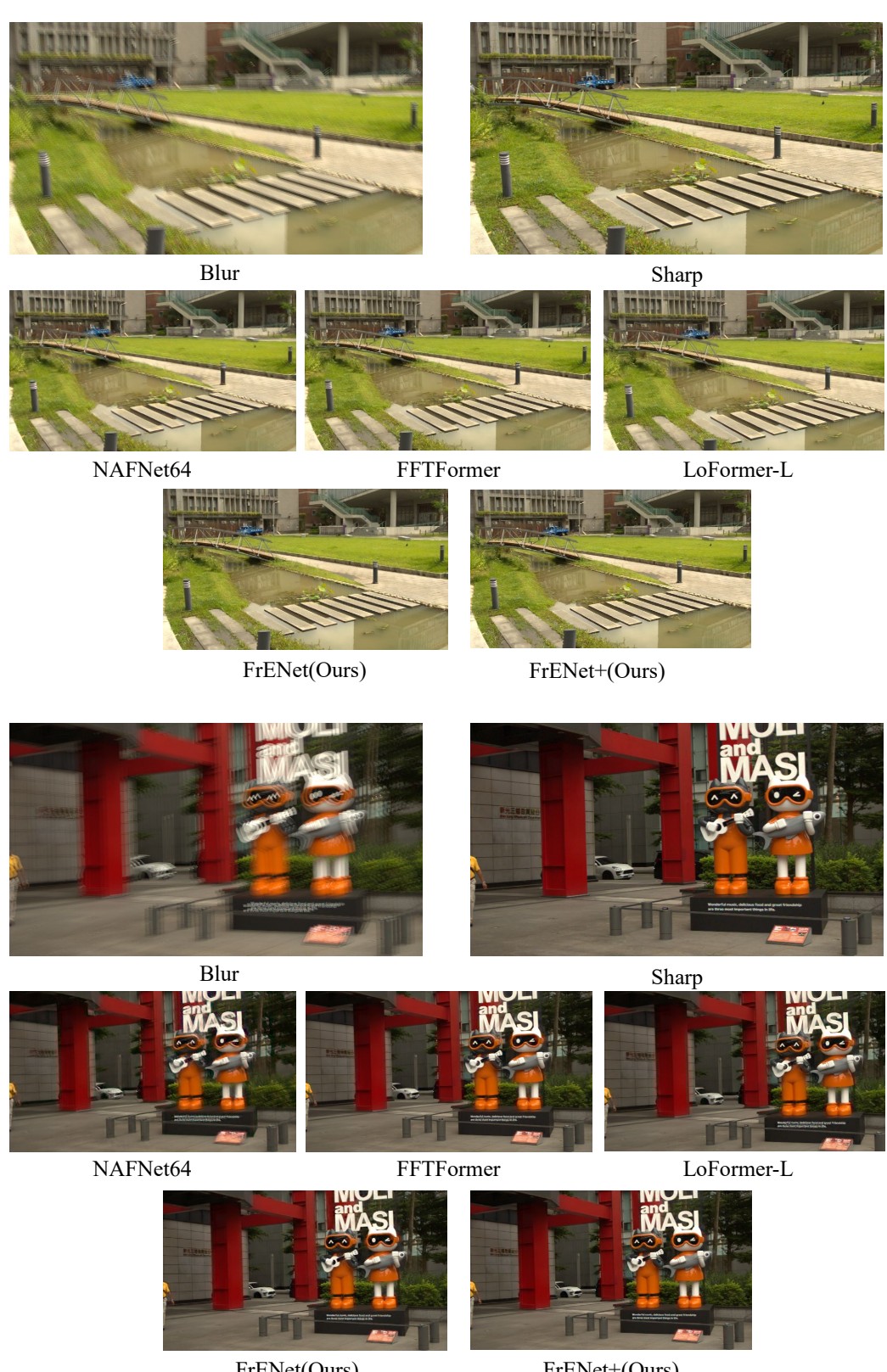

Figure D.5: Visual results on the Deblur-RAW dataset.

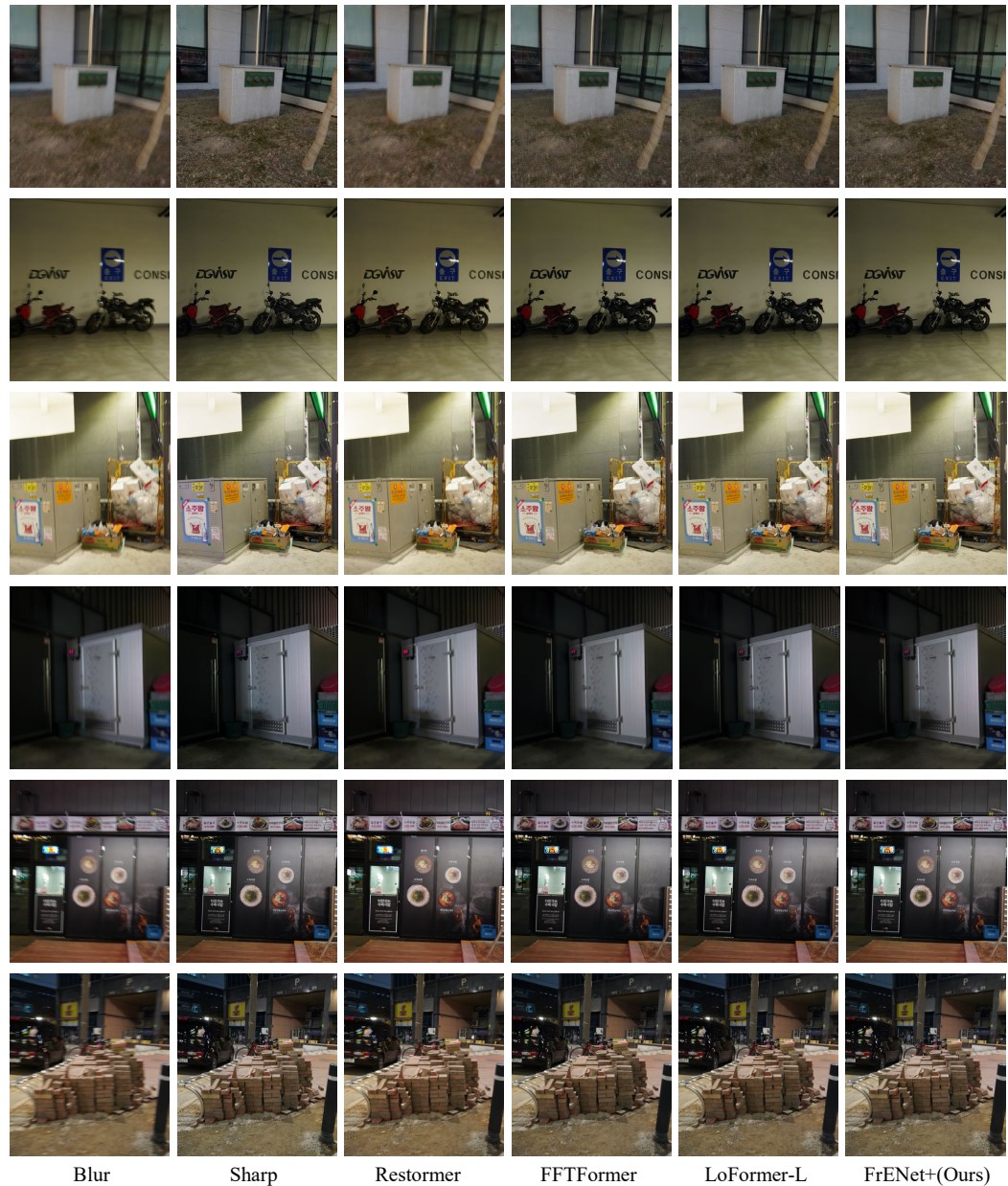

| Blur | Sharp | Restormer | FFTFormer | LoFormer-L | FrENet+(Ours) |

Figure D.6: Visual results on the RealBlur-J dataset.

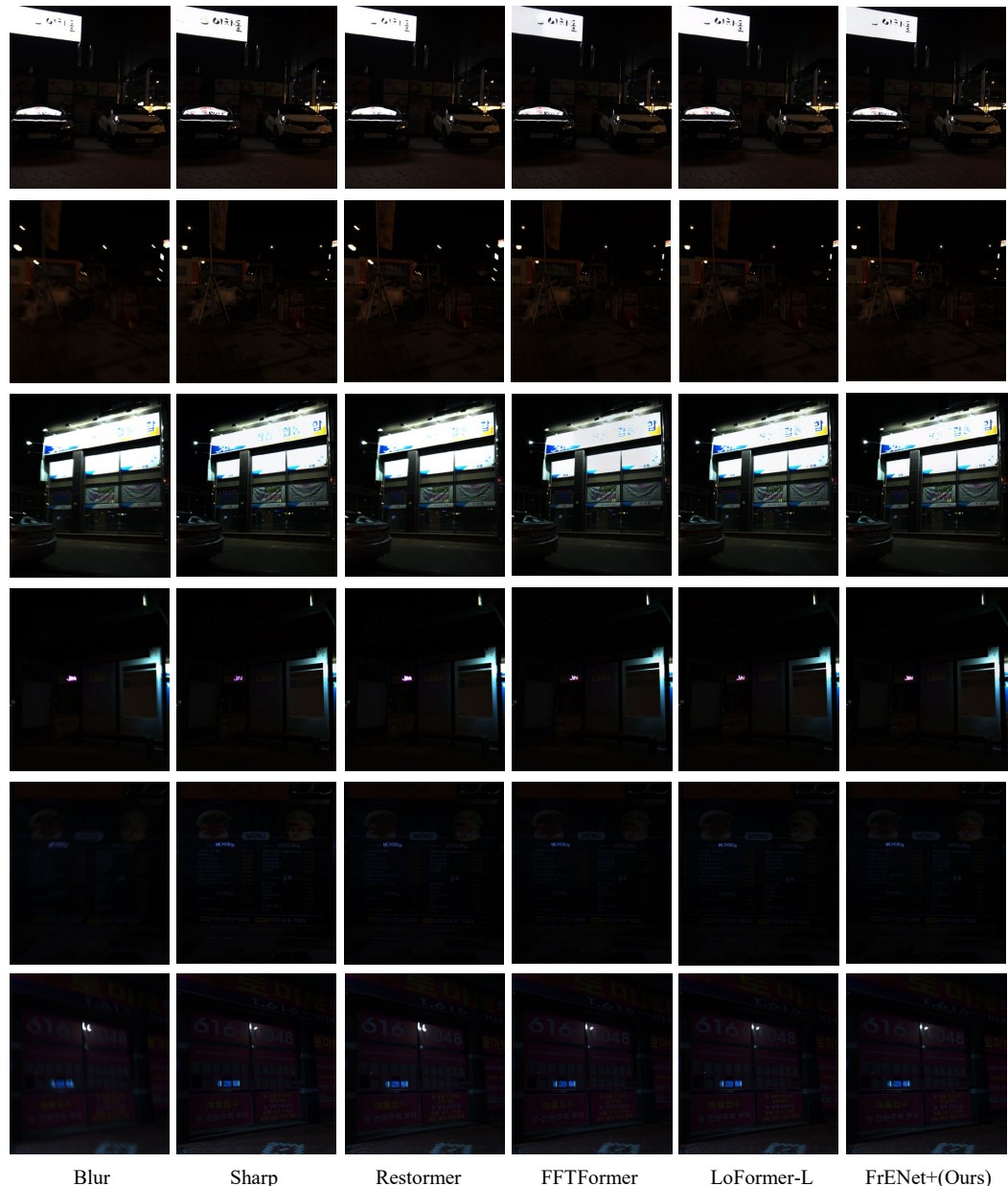

| Blur | Sharp | Restormer | FFTFormer | LoFormer-L | FrENet+(Ours) |

Figure D.7: Visual results on the RealBlur-R dataset.

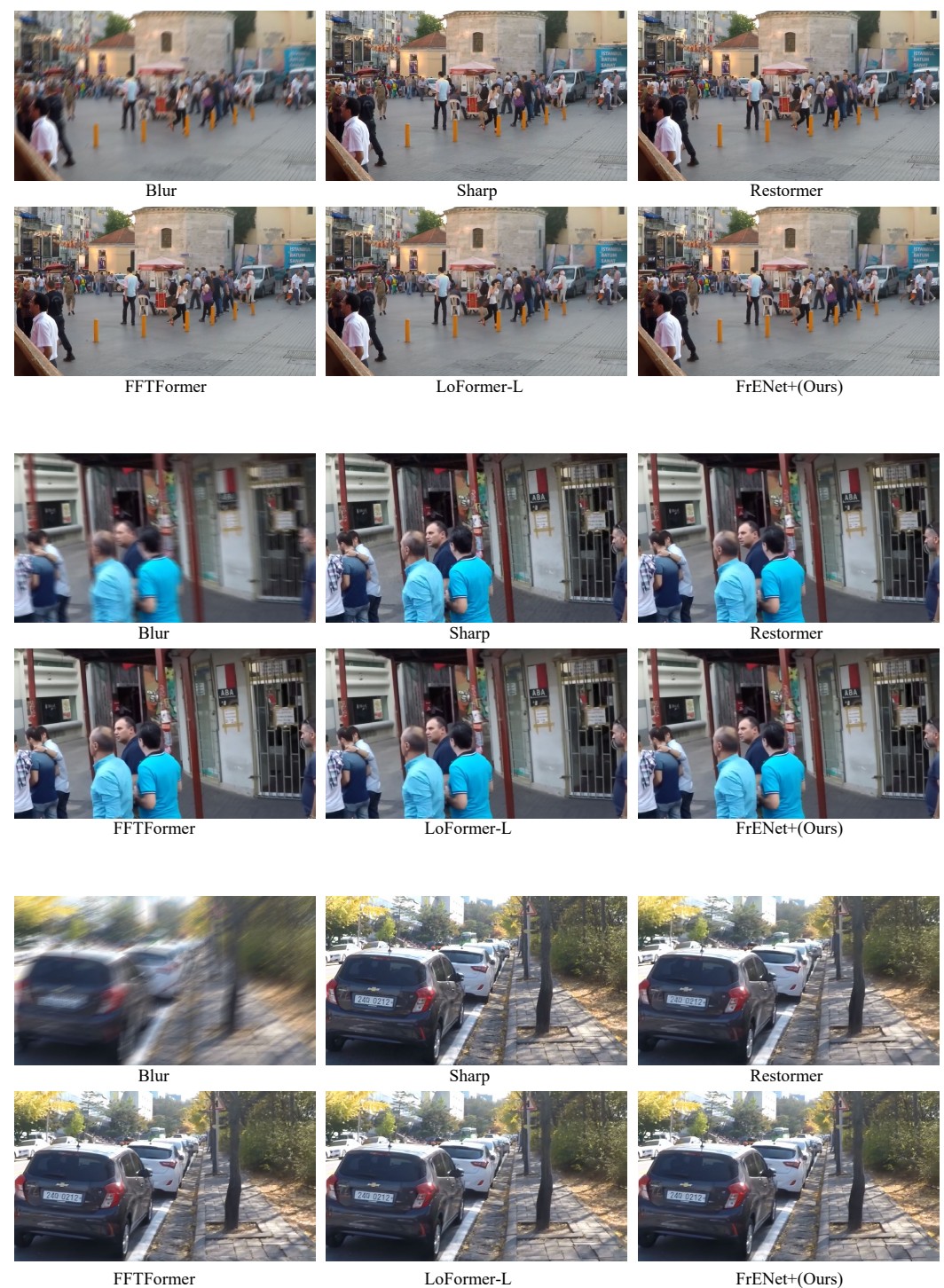

Figure D.8: Visual results on the GoPro dataset.

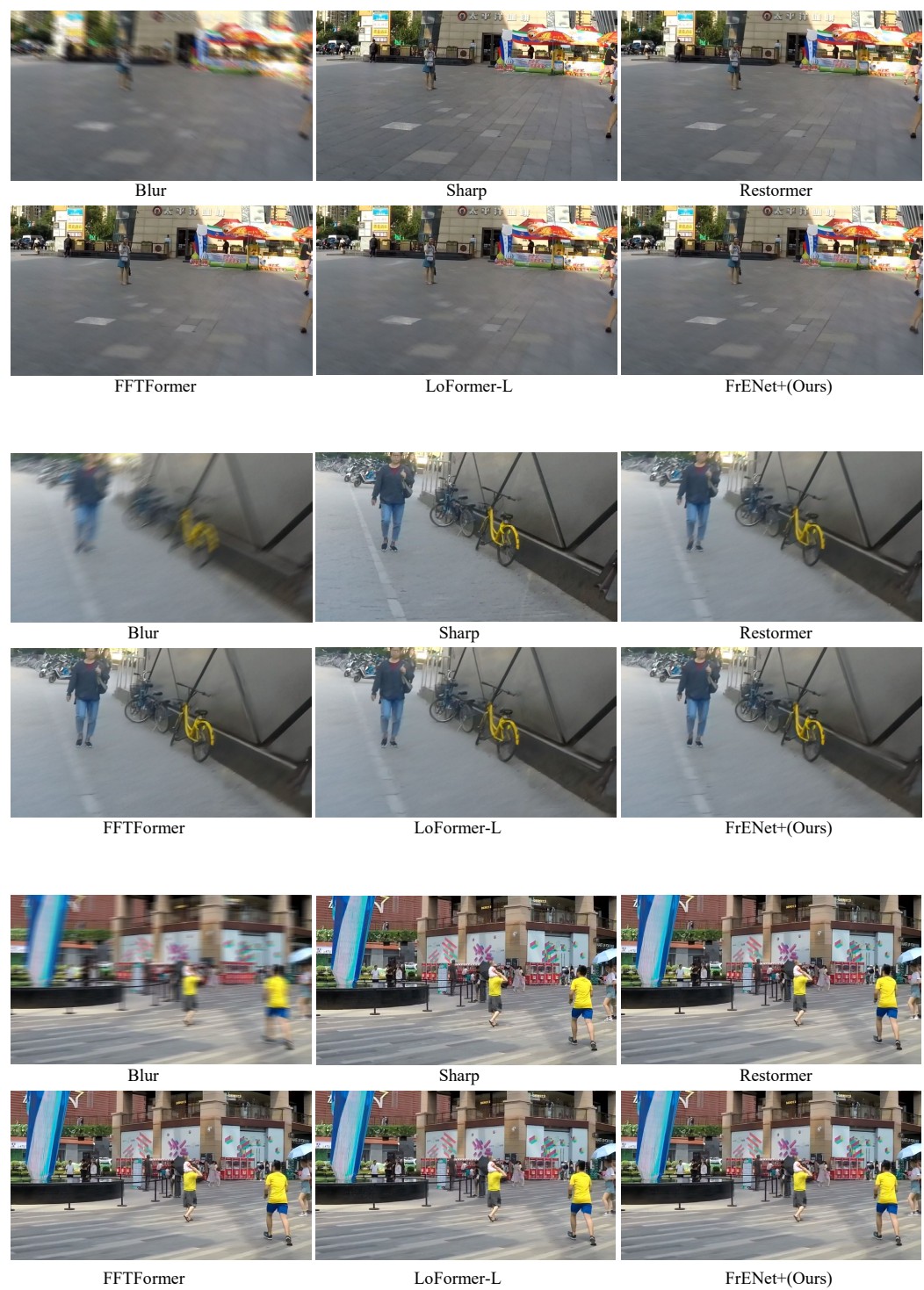

Figure D.9: Visual results on the HIDE dataset.

# E Boder Impacts

While advanced deblurring algorithms offer simple image enhancement to the public, their use also raises concerns about potential malicious applications, especially regarding privacy issues. Blurring is often used to protect personal information, such as faces and personal IDs. To prevent potential misuse, image forensics algorithms can be used, which are designed to authenticate images. Many of these algorithms focus on training classifiers to distinguish between images captured in the real world and images processed by deep learning models.

