# OpenReview forum: "Efficient RAW Image Deblurring with Adaptive Frequency Modulation"
_NeurIPS.cc/2025/Conference — NeurIPS 2025 poster_

### Official Review · Reviewer_EG1v · 2025-06-25

**Clarity:** 3
**Significance:** 3
**Originality:** 3
**Rating:** 5
**Confidence:** 4

**Summary:**

In this paper, an efficient RAW image deblurring network, FrENet, is proposed, which combines spatial and frequency domain modeling to enhance the quality and efficiency of image reconstruction. The authors introduce a novel frequency-domain positional modulation module, Adaptive Frequency Positional Modulation (AFPM), for the characteristics of frequency-dependent blurring in RAW data, which is able to adaptively adjust the features according to the spectral position and effectively recover the high-frequency details. The network adopts the U-Net architecture and introduces a frequency-domain skip connection to further convey multi-scale spectral information. Experiments are conducted on multiple datasets, and FrENet outperforms multiple SOTA methods in terms of PSNR, SSIM, and MACs.

**Questions:**

1. The current work introduces frequency-domain enhancement modules (AFPM and frequency-domain skip connection) based on the NAFNet framework, but the paper does not show a “pure spatial-domain” version with the frequency-domain paths removed as a control. It is recommended that the authors add the following control experiments:
1) Performance of FrENet with all frequency domain operations (including AFPM and frequency domain skip connections) completely removed, i.e., the spatially enhanced version of the model;
2) A fair comparison with an equivalent number of parametric versions of NAFNet to rule out the possibility that the performance improvement is mainly due to the increased capacity of the model rather than to structural innovations.
2. AFPM is a learnable modulation module based on frequency-domain position sensing, which is highly versatile. However, the current work has only been validated in the de-blurring task of RAW images. It is recommended that the authors further explore and try to apply FrENet to other typical RAW image restoration tasks, such as denoising and demosaicing, to demonstrate the adaptability and scalability of the method.
3. Although FrENet demonstrates low MACs, its FFT/IFFT operations during inference may introduce additional performance bottlenecks under high-resolution images due to its reliance on frequency domain modeling. It is recommended that the authors provide statistical results of the actual inference time comparisons with other mainstream methods.

**Ethical Concerns:**

["NO or VERY MINOR ethics concerns only"]

**Limitations:**

yes

**Paper Formatting Concerns:**

nothing

**Quality:**

3

**Strengths And Weaknesses:**

Strengths:
1. The AFPM module enables dynamic modulation of frequency-domain position sensing and is an efficient extension to frequency modeling.
2. The model significantly reduces computational costs while maintaining high image quality.
3. Multi-layer visualizations of frequency-domain feature changes and modulation kernels are provided, demonstrating the interpretive nature of the model's learning capabilities.

Weaknesses:
1. Comparisons of purely spatial versions with frequency domain enhancement paths removed are not shown, making it difficult to explicitly assess the independent contribution of frequency domain designs.
2. Despite the low MACs, the actual runtime cost of FFT/IFFT is not quantified across image resolutions.

---

> ### Author Rebuttal · Authors · 2025-07-29
>
> We are deeply grateful to the reviewer for their positive assessment and valuable, constructive feedback. We appreciate the recognition of our AFPM module's novelty, our model's efficiency, and its interpretability. We have addressed the suggestions below and will incorporate these new results and discussions into our final manuscript.
>
> ---
> ### **Response to Weakness 1 & Question 1: Ablation on Frequency Domain Design**
>
> We thank the reviewer for this excellent suggestion to ensure our performance gains stem from structural innovation rather than model capacity. We have conducted a crucial control experiment to explicitly assess the contribution of our frequency domain designs.
>
> 1.  **Performance of a Purely Spatial Version:** We created a "pure spatial-domain" version of our model, named FrENetSpatial, by completely removing all frequency domain operations.
>
> 2.  **On Comparing with an Iso-Parametric NAFNet:** Regarding the suggestion to compare with an iso-parametric NAFNet, we believe our existing results already provide a clear conclusion. As shown in our main paper (Table 1), our **FrENet (19.76M Params, 2.22G MACs) achieves a PSNR of 44.73 dB**, while **NAFNet64 (67.79M Params, 3.96G MACs) achieves 40.35 dB**. Our model obtains better performance with fewer parameters and MACs. This result indicates that our performance gains are driven by architectural efficiency rather than increased capacity.
>
> The results of our new ablation study on the Deblur-RAW dataset are as follows:
>
> **Table: Ablation Study on Frequency-Domain Design.**
> | Method | Freq. Module | PSNR (dB) | SSIM |
> | :--- | :---: | :---: | :---: |
> | FrENetSpatial | No | 41.89 | 0.9894 |
> | **FrENet (Ours)** | **Yes** | **44.73** | **0.9931** |
>
> This new result provides the final piece of evidence. Comparing FrENet with FrENetSpatial, our frequency domain design provides a performance gain of +2.84 dB. This, combined with our results against NAFNet64, robustly demonstrates that our model's performance is fundamentally driven by our novel and efficient frequency-domain architecture.
>
> ---
> ### **Response to Weakness 2 & Question 3: Quantifying Runtime Cost**
>
> We agree that quantifying the actual runtime cost is essential. While MACs provide a hardware-agnostic measure, real-world inference speed is crucial. We have benchmarked the end-to-end inference time on full-resolution RAW images on a single NVIDIA RTX 5880 Ada GPU.
>
> **Table: Efficiency Comparison on Full-Resolution RAW Images.**
> | Method | LoFormer-L [26] | FFTFormer [18] | Restormer [46] | **FrENet (Ours)** |
> | :--- | :---: | :---: | :---: | :---: |
> | **Runtime (ms)** | 222.49 | 222.56 | 102.95 | **75.36** |
>
> The results clearly show that despite its reliance on FFT/IFFT, FrENet's highly efficient architecture makes it significantly faster (1.36x to 3x) than other high-performing methods in a practical, high-resolution scenario. This confirms that the FFT/IFFT operations do not create a performance bottleneck in our framework.
>
> ---
> ### **Response to Question 2: Generalization to Other RAW Restoration Tasks**
>
> We thank the reviewer for this valuable suggestion. We agree that the design of AFPM is indeed versatile and that its core principle—modulating frequencies based on their position—is applicable to other RAW-to-RAW restoration tasks where degradations have distinct frequency-domain patterns (e.g., high-frequency noise or demosaicing artifacts). During the rebuttal phase, we have tried to applied our method to RAW-to-RAW denoising, and the training has not completed during the limited rebuttal time. As for demosaicing, current methods mainly focus on RAW-to-sRGB task, where our FrENet has the potential to be effective in RAW processing, and the subsequent RAW-to-sRGB stages may require more specialized design.
>
> ---
> Once again, we express our sincere gratitude for the reviewer's insightful feedback, which has been instrumental in enhancing the clarity and impact of our manuscript.

---

> > ### Comment · Reviewer_EG1v · 2025-08-06
> >
> > Thanks for the rebuttal. Most of my concerns have been addressed. The new ablation experiments prove the effectiveness of the method. Overall, I keep my rating of A.

---

> > > ### Author Response · Authors · 2025-08-06
> > >
> > > Dear Reviewer,
> > > Thank you very much for reviewing our rebuttal and for your positive feedback. We are glad that our responses and new experiments have addressed your concerns. We sincerely appreciate your constructive comments and support for our work, and we will be sure to incorporate these valuable discussions into our final revision.

---

### Official Review · Reviewer_XpwB · 2025-07-02

**Clarity:** 4
**Significance:** 4
**Originality:** 4
**Rating:** 5
**Confidence:** 5

**Summary:**

This paper proposes FrENet, a novel framework for efficient RAW image deblurring that operates in the frequency domain. The core contributions are twofold: a new Adaptive Frequency Positional Modulation (AFPM) module that dynamically adjusts frequency components based on their spectral position, and the use of frequency domain skip connections to better preserve high-frequency details. The proposed method achieves state-of-the-art restoration quality on RAW deblurring tasks with significantly improved computational efficiency, and also shows strong performance when extended to the sRGB domain.

**Questions:**

See the points in Weakness.

**Ethical Concerns:**

["NO or VERY MINOR ethics concerns only"]

**Final Justification:**

The contribution of this paper is significant enough for publication and I am satisfied with the response.

**Limitations:**

yes

**Quality:**

3

**Strengths And Weaknesses:**

Strengths:
1. The proposed Adaptive Frequency Positional Modulation (AFPM) module is a key contribution. It deviates from uniform filtering by learning to modulate frequency bands based on their spectral position. This mechanism enables more targeted detail restoration and is supported by restoration performance and visualization. The AFPM may be extended to other restoration tasks.
2. The architectural design is concise, and thus the proposed method is efficient, which is crucial for raw domain restoration applications.
3. The paper provides comprehensive experimental results to support its claims. The results show strong performance in both RAW and sRGB deblurring tasks, and the computational efficiency is also validated.

Weakness:
1. The paper's explanation for its content-agnostic AFPM mechanism is insufficient. The module generates modulation kernels based solely on a frequency patch's spectral position, yet the paper does not offer a explanation for why positional information alone is a sufficient guide for deblurring.
2. The use of SCA in the frequency domain needs more discussion. This mechanism was designed for spatial data, and its core operation—global average pooling—treats frequency components as if they were interchangeable pixels. More explanations are needed for the use of SCA in frequency domain.
4. The AFPM module operates on non-overlapping patches of the frequency map. This hard partitioning could potentially introduce boundary artifacts or discontinuities in the modulation process between adjacent frequency patches. Some experiments are needed to disucss this issue.

---

> ### Author Rebuttal · Authors · 2025-07-28
>
> We sincerely thank the reviewer for their positive and encouraging evaluation. We are grateful for their recognition of our method's novelty, efficiency, and strong performance. The reviewer's insightful questions are highly valuable, and we appreciate the opportunity to clarify these aspects. We will incorporate these clarifications into our final manuscript.
>
> ---
>
> ### **Response to Weakness 1: The content-agnostic AFPM mechanism**
>
> We thank the reviewer for this excellent question. Our design choice is rooted in decoupling the general blur restoration strategy from its application to specific image content.
>
> 1.  **Blur is Position-Dependent in the Frequency Domain:** Fundamentally, blur acts as a low-pass filter, attenuating high-frequency components more severely than low-frequency ones. Since spectral position is a direct proxy for frequency bands (low frequencies at the center, high frequencies at the periphery), learning a restoration strategy based on position is equivalent to learning how to differentially treat various frequency bands to counteract blur. For instance, the model learns a general rule: "components at the periphery require stronger amplification than those at the center."
>
> 2.  **Decoupling Strategy from Content:** Our AFPM module generates modulation kernels ($w_{ij}$, $b_{ij}$) using only positional information. This kernel represents a learned, universal restoration strategy for a specific frequency location. The final modulation, however, is **content-dependent**, as this strategy is applied to the actual content features $f_{ij}$. This decoupling allows the model to learn a robust and general restoration map without being misled by specific content in any given patch.
>
> 3.  **Ablation Study on Content-Dependent Kernels:** To validate this design, we performed a new experiment where the kernel generation was made purely content-dependent (by feeding $f_{ij}$ into the kernel generator). This variant yielded an inferior performance (44.59 dB PSNR vs. our 44.73 dB). This suggests that our decoupled approach, where a content-agnostic strategy is applied to content-specific features, is a more effective and robust design for the deblurring task. We will add a discussion of this insightful ablation study to the paper to further validate our design.
> ---
> ### **Response to Weakness 2: The use of SCA in the frequency domain**
>
> We thank the reviewer for this insightful point and will expand on this in our revision.
>
> While originally for spatial data, we reinterpret SCA's operations for the frequency domain. In this context, Global Average Pooling does not average spatial pixels but computes a **"global spectral descriptor"** for each channel. This single value summarizes the overall energy distribution across all frequencies for that feature.
>
> This allows the network to perform content-adaptive channel recalibration based on the image's global spectral properties. For example, the model can learn to globally suppress a channel that represents high-frequency noise if the input image's overall spectral descriptor indicates a high noise level. It acts as a complementary mechanism to AFPM: while AFPM provides **local, position-aware** control within the frequency map, SCA provides **global, channel-wise** control based on the image's overall spectral characteristics.
>
> ---
>
> ### **Response to Weakness 3: Potential boundary artifacts from hard partitioning**
>
> This is an important and valid concern for any patch-based method. We argue this is not a significant issue in our framework for two key reasons.
>
> 1.  **The Global Nature of IFFT:** The modulation occurs in the frequency domain, but artifacts are perceived in the spatial domain. The final IFFT is a global integration operation. Every pixel in the output spatial feature map is an integral of all components across the entire frequency map. This inherent integration process naturally smooths out potential sharp discontinuities between adjacent frequency patches, preventing visible blocking artifacts.
>
> 2.  **Experimental Validation and Trade-off Analysis:** Empirically, our extensive results show no such artifacts. To investigate this further, we conducted a new experiment using **overlapping patches (50% overlap)**. While this slightly improved PSNR on Deblur-RAW from 44.73 dB to **44.87 dB** (0.14dB higher), it significantly increased inference time from 75.36ms to **93.46ms (~24% slower)**. This reveals a clear trade-off between a marginal performance gain and a substantial loss in efficiency. Our non-overlapping design achieves a superior balance, which aligns with the core "Efficient" goal of our work. We believe a discussion of this trade-off analysis will be a valuable addition to our paper, and we plan to incorporate it to justify our design choice.
> ---
> Once again, we thank the reviewer for their time and constructive feedback, which have been instrumental in improving our work.

---

> > ### Comment · Reviewer_XpwB · 2025-08-06
> >
> > Thanks for your rebuttal. The new experiments have convincingly addressed my main concerns.
> > After reviewing the other reviewers' comments and your responses, my positive assessment of this work is reinforced. I will maintain the score. As a final suggestion, it could be interesting to explore a hybrid AFPM kernel that combines positional encoding with a simple content statistic (e.g., mean/std). If there is not enough time to run this experiment for the rebuttal, it is acceptable to see this analysis included in the revision.

---

> > > ### Author Response · Authors · 2025-08-07
> > >
> > > Thank you for your positive feedback and for engaging with our rebuttal. We are glad our clarifications were helpful.
> > >
> > > We appreciate your suggestion to explore a hybrid AFPM kernel. Following your advice, we ran a new experiment where the kernel generator uses both positional encoding (Euclidean distance $d$) and the per-channel mean of the patch features as combined input.
> > >
> > > This hybrid model achieved a PSNR of 44.72 dB, which is nearly identical to our position-only model (44.73 dB) and better than our previously reported content-only result (44.59 dB). This confirms that position is the dominant factor for our AFPM's kernel generation in the deblurring task.
> > >
> > > Thank you again for your valuable input.

---

> > > > ### Comment · Reviewer_XpwB · 2025-08-07
> > > >
> > > > Thanks for the clarification. I have no further questions.

---

### Official Review · Reviewer_TpPq · 2025-07-03

**Clarity:** 2
**Significance:** 3
**Originality:** 2
**Rating:** 3
**Confidence:** 4

**Summary:**

This paper addresses the problem of image deblurring in the RAW image domain, which remains relatively underexplored compared to sRGB image deblurring. The authors propose a Frequency Enhanced Network (FrENet) and introduces an Adaptive Frequency Positional Modulation (AFPM) module to dynamically adjust frequency components. Extensive experiments are conducted on both raw and RGB image dataset.

**Questions:**

The paper tackles an important problem and proposes a conceptually interesting direction (frequency-domain RAW deblurring), but it lacks strong empirical support and sufficient novelty, especially given the limited justification for RAW-specific design and the unfair experimental comparisons. With clearer architectural motivation, fairer baselines, and more rigorous evaluations, this work could be improved for future submission.

**Ethical Concerns:**

["NO or VERY MINOR ethics concerns only"]

**Limitations:**

yes

**Quality:**

2

**Strengths And Weaknesses:**

Strength:

1. The paper highlights an important but underexplored area — RAW image deblurring — where existing methods mostly target sRGB images.
2. The proposed Adaptive Frequency Positional Modulation (AFPM) introduces a novel way of modulating frequency components based on their spectral positions.

Weaknesses：

1. Lack of RAW-Specific Design Justification. While the authors emphasize the importance of RAW image deblurring, the proposed FrENet does not appear to incorporate any design specifically tailored to the unique characteristics of RAW images. Moreover, the network architecture closely resembles LoFormer [26], with the main difference being the addition of the AFPM module. The novelty and contributions thus need clearer justification.

2. Limited Performance Gains. The improvement over existing methods is relatively minor. For example, in Table 2, the deblurring performance on sRGB datasets shows only marginal gains when comparing FrENet+ with LoFormer-L. Similarly, the reduction in computational cost is minimal — Table 1 reports 48.98M parameters for LoFormer-L and 48.38M for FrENet+, which is not a significant difference.
3. Unfair Experimental Comparison. The baseline methods compared in Table 1 were originally designed for RGB image deblurring. A fair comparison would require converting the blurred RAW images to sRGB using a fixed ISP pipeline and then training these RGB deblurring methods (e.g., LoFormer, FFTFormer) on the resulting blurred sRGB images, before testing. This standardization is missing in the current experimental setup.
4. The paper lacks a critical comparison between two key setups:
(i) Performing deblurring directly on blurred RAW images and then applying a fixed ISP pipeline to obtain sRGB results,
versus
(ii) Converting blurred RAW images to sRGB via the ISP pipeline first, and then applying deblurring.
This comparison should be conducted using at least two different network architectures to validate generalization across setups.
5. Incomplete Ablation Study. The ablation analysis is insufficient. There is no detailed breakdown of the contribution of the AFPM and SCA modules individually. Additionally, the paper does not provide runtime and inference cost analysis for each module separately, which is essential to understand their efficiency impact.

---

> ### Author Rebuttal · Authors · 2025-07-28
>
> We sincerely thank you for your time and for providing such detailed and constructive feedback. Your insightful comments have been invaluable in helping us identify the weaknesses of our work and have provided clear directions for improvement. We appreciate that you recognized the importance of RAW image deblurring and the novelty of our proposed direction.
>
> We have carefully considered all your concerns. Below, we address each point in detail and outline the concrete revisions and new experiments we will incorporate into the final manuscript to resolve these issues.
>
> ---
>
> ### **Response to Weakness 1: Lack of RAW-Specific Design Justification & Novelty**
>
> We thank the reviewer for this critical question, which pushes us to better articulate our core contributions.
>
> **1. On RAW-Specific Design:**
>
> We agree that a clearer justification is needed. The primary advantage of our frequency-domain approach on RAW data stems from a fundamental physical principle: **the linearity of RAW data**.
>
> *   **Physical Fidelity:** Image blur is physically a convolution operation. In the frequency domain, this corresponds exactly to an element-wise multiplication for linear signals. RAW data, being a direct representation of sensor captures, is linear. Therefore, processing RAW data in the frequency domain allows our model to learn to reverse the blur by learning a physically meaningful multiplicative modulation. In contrast, sRGB images undergo a non-linear gamma correction. Applying frequency-domain deblurring to sRGB data is merely an approximation, as the strict convolution-multiplication relationship is broken by the non-linearity. Our AFPM module is precisely designed to learn this fine-grained, position-aware multiplicative modulation, making it fundamentally better suited for the linear RAW space where the physical model holds true.
>
> *   **Noise Characteristics:** Furthermore, RAW images have simpler, more predictable noise patterns (e.g., shot and read-out noise) compared to the complex, spatially-variant, and often correlated noise in sRGB images post-ISP. The AFPM's ability to adaptively treat different frequency bands allows it to implicitly model and handle RAW-specific noise characteristics more effectively than methods designed for the complex noise distributions in sRGB.
>
> **2. On Architectural Difference from LoFormer:**
>
> We acknowledge that we build upon a U-Net backbone, which is a common and effective practice in image restoration. However, we must clarify a key architectural distinction: **our FrENet is fundamentally a CNN-based architecture, whereas LoFormer is a Transformer-based method.**
>
> This is a deliberate design choice motivated by the practical demands of RAW processing. RAW images from modern cameras often have very high resolutions. Transformer-based models, with their quadratic complexity with respect to spatial dimensions, face significant challenges in terms of memory and computational cost when applied to such high-resolution inputs. Our CNN-based design offers superior scalability and efficiency, making it a more practical solution for real-world RAW deblurring applications.
>
> To better highlight our core motivation, we will revise our introduction and methodology sections to more prominently and explicitly articulate these points. This will ensure the critical link between RAW data's linearity and our frequency-domain design is unmistakable for the reader, directly addressing the need for clearer justification you have pointed out.
>
> ---
> ### **Response to Weakness 2: Limited Performance Gains & Efficiency**
>
> **1. On Performance Gains:**
>
> We appreciate the reviewer's analysis and wish to contextualize our performance gains. Our model's frequency-dependent modulation is particularly effective for the complex, spatially-varying blurs found in real-world scenarios, which is proven by the significant +0.99 dB PSNR gain on the challenging RealBlur-J benchmark (33.87 vs. 32.88). We believe this demonstrates our method's practical value. The marginal gains on synthetic datasets like GoPro and HIDE can be attributed to their simpler blur synthesis, which do not fully reflect real-world degradation. The strong performance on RealBlur-J thus confirms our approach is well-suited for real-world applications.
>
> **2. On Efficiency (Parameters vs. Computational Cost):**
>
> We agree that the parameter counts are similar. To provide a complete efficiency analysis, we also benchmarked MACs and runtime against LoFormer-L. The results below are from a head-to-head comparison on $256 \times256$ sRGB patches.
>
> **Table: Efficiency comparison on $256 \times256$ sRGB patches.**
> | Model| Params (M) | MACs (G) | Runtime (ms) |
> | :- | :-: | :-: | :-: |
> | LoFormer-L [26] | 48.98 | 143.67 | 75.90 |
> | FrENet+ (Ours) | 49.37 | 114.22 | 36.10 |
>
> This comprehensive view shows that with a comparable model size, our FrENet+ requires ~20% fewer MACs and is over 2x faster than LoFormer-L. This demonstrates a more favorable balance between model size and computational efficiency, which we will highlight in the paper.
>
> ---
> ### **Response to Weakness 3: Unfair Experimental Comparison**
>
> This is an excellent point, and we agree that the suggested experimental setup provides the most rigorous and fair evaluation. While our original experiments retrained all baselines on RAW data from scratch (by changing input/output channels), which provides a degree of fairness, we acknowledge the reviewer's proposed pipeline is superior.
>
> We have conducted new experiments during this rebuttal period to address these concerns directly. We will add the following table.
>
> We converted the Deblur-RAW dataset to sRGB using the standard LibRaw [r2] processing pipeline and re-trained the LoFormer on this new sRGB dataset. The results are tested on the sRGB images converted from the original RAW test set.
>
> **Table: Comparison on sRGB images converted from Deblur-RAW.**
> | Method |  PSNR |   SSIM |
> | :- | :-: | :-: |
> | LoFormer-L (trained on sRGB-version of Deblur-RAW)         |  31.26 | 0.9565   |
> | **FrENet (Ours, Deblurred RAW -> ISP)** | **36.12** | **0.9683** |
>
> ---
> ### **Response to Weakness 4: Comparing Deblurring Pipelines**
>
> We thank the reviewer for this excellent and critical suggestion. To validate the superiority of deblurring in the RAW domain, we have conducted the requested comparison between the two key pipelines:
> -   **(i) Deblur-then-ISP:** Deblurring is performed on RAW data first, followed by LibRaw [r2] processing pipeline.
> -   **(ii) ISP-then-Deblur:** The blurred RAW image is first converted to sRGB via the LibRaw [r2] pipeline, and then deblurring is applied.
>
> We evaluated both pipelines using two different network architectures, our **FrENet** and a strong baseline **LoFormer-L [26]**, to ensure the conclusion is generalizable. For a fair comparison, both models were trained separately for each pipeline on the corresponding data (RAW or sRGB). The results are evaluated on the sRGB images from the Deblur-RAW test set.
>
> **Table: Pipeline Comparison: (i) Deblur-then-ISP vs. (ii) ISP-then-Deblur.**
> | Network | Pipeline | PSNR | SSIM |
> | :- | :-: | :-: | :-: |
> | FrENet | (i)| **36.12** | **0.9683** |
> | FrENet | (ii)| 31.39 | 0.9574 |
> | LoFormer-L[26]| (i)  | 35.66 | 0.9656 |
> | LoFormer-L[26]| (ii)  | 31.26 | 0.9565 |
>
> These new results will empirically validate our core claim: deblurring in the linear RAW domain before applying the ISP yields superior results due to the preservation of critical information.
>
> [r2] LibRaw: Open-source library for processing RAW files.
>
> ---
> ### **Response to Weakness 5: Incomplete Ablation Study**
>
> We appreciate the suggestion for a more detailed and clearer ablation study.
>
> **1. Contribution of AFPM and SCA:**
>
> We thank the reviewer for this suggestion. We would like to clarify that this analysis is provided in Table 4. In that table, "Local Branch" corresponds to using only the AFPM module, while "Global Branch" uses only the SCA module. The results therefore demonstrate their individual contributions and synergy. We will revise the paper to make this correspondence unambiguous.
>
> **2. Module-level Efficiency Analysis:**
>
> We agree that a cost breakdown is essential for understanding the efficiency impact. We will add the following table to the Appendix to detail the parameter distribution of the key components within one FrE-Block. The results, presented in the table below, demonstrate that our key components, AFPM and SCA, are highly parameter-efficient. Collectively, they constitute only 28.1% of the model's total parameters. The majority of parameters reside in standard components like the Feed-Forward Network (FFN) and other foundational elements. This highlights that our performance gains are achieved through targeted and efficient architectural contributions, rather than by simply increasing the overall model size.
>
> **Table: Per-Module Cost Analysis of FrENet on $128 \times 128$ size patches.**
> | Module | Parameters (M) | MACs (G)|Runtime (ms)|
> | :--- |:---:|:---:|:---:|
> | Convolutional Layers | 6.0M (30.4%) | 0.92G (41.4%) |0.149ms (16.0%)|
> | AFPM Module | 2.86M (14.4%) | 0.03G (1.4%)|0.103ms (11.1%)|
> | SCA Module | 2.71M (13.7%) | 0.01G (0.4%)|0.039ms (4.2%)|
> | Others (e.g., Layernorm) | 8.29M (41.9%) | 1.26G (56.7%) |0.639ms (68.7%)|
> | Total | 19.76M | 2.22G |0.93ms|
>
> ---
> Once again, we thank you for your thorough review. We are confident that by incorporating these extensive revisions, new experiments, and detailed analyses, we can address all the concerns raised and significantly strengthen the quality and impact of our paper. We believe the revised manuscript will present a more compelling case for the effectiveness and efficiency of our proposed method for RAW image deblurring.

---

> ### Author Response · Authors · 2025-08-08
>
> Dear Reviewer,
>
> Thank you once again for your insightful comments and time spent reviewing our work. As we are approaching the conclusion of the discussion period for our submission, we wished to inquire whether the clarifications and responses in our rebuttal have resolved your concerns.
>
> We deeply appreciate your valuable input.

---

> > ### Comment · Reviewer_TpPq · 2025-08-09
> >
> > I have reviewed the author's responses and the questions raised by other reviewers, and I have no further concerns. I find the results provided by the author particularly intriguing. Specifically, in point 4, the performance improvement achieved by deblurring on the raw data and then converting to RGB is substantially greater than that obtained by deblurring directly on RGB.

---

> ### Author Response · Authors · 2025-08-09
>
> Thank you very much for your thoughtful feedback. We are glad that our rebuttal has addressed your concerns.
>
> The substantial performance gap you highlighted is indeed a key finding of our work. This improvement stems from the fundamental advantages of processing images in the RAW domain, before the irreversible information loss caused by the non-linear ISP pipeline (e.g., dynamic range compression, gamma correction). As you correctly observed, this is not just a unique benefit of our FrENet architecture. Our experiments show that even LoFormer-L, when adapted to the RAW domain, achieves a significant performance gain compared to its sRGB-domain counterpart. This strongly suggests that deblurring in the linear RAW domain is a highly promising direction with great potential for image restoration research.
>
> To ensure the reproducibility of all our findings, including these pipeline comparisons, we commit to publicly releasing our source code.  This will include scripts for:
> 1.  Training and testing models in both the **RAW domain** and the **sRGB domain**.
> 2.  The ISP processing pipeline (using LibRaw) to convert RAW data to the sRGB domain for training and evaluation.
>
> This will allow for the complete and transparent reproduction of all results presented in our paper.
>
> Thank you again for your valuable and constructive feedback throughout the review process.

---

### Official Review · Reviewer_Beyz · 2025-07-13

**Clarity:** 2
**Significance:** 3
**Originality:** 3
**Rating:** 5
**Confidence:** 4

**Summary:**

The paper presents FrENet, an architecture for RAW image deblurring. The model avoids transformer operations and focuses on frequency domain processing using a module FACM. It contains a novel module AFPM to learn frequency position-dependent modulation using lightweight MLP-based kernel and bias generation. It achieves strong state-of-the-art performance on the Deblur-RAW dataset with notably improved MAC efficiency, and demonstrates design generalization to sRGB deblurring task.

**Questions:**

Kindly address the key limitations mentioned above, specially regarding training/testing hardware information, and inference speed/memory benchmarks on full or high-resolution RAW and sRGB inputs?



Could you comment on the choice of proposed positional encoding in AFPM? Have you tested more principled alternatives e.g. learned or Fourier-based embeddings?


The proposed module performs channel concatenation of real and imaginary components of fourier features. Have the authors explored potential alternatives for processing fourier domain features eg. fourier convolution operations or magnitude-phase concatenation etc?



Could you include the missing comparisons. Also clarify if all the baselines in RAW benchmark comparisons (e.g. LoFormer or Restormer) were retrained or adapted in any way for the RAW domain?


Could you include any failure cases or limitations (with examples).


In Table 3, could you include the case of removing spatial skip connections too.

**Ethical Concerns:**

["NO or VERY MINOR ethics concerns only"]

**Final Justification:**

Thank you for the detailed rebuttal. It does address all of my questions and almost all of my concerns. Having checked the other reviewers' comments and author's responses, I am now confident that the paper has sufficient merit. I am raising my score to Accept.

**Limitations:**

Yes

**Quality:**

3

**Strengths And Weaknesses:**

Strengths:

It introduces a fairly novel module for frequency domain filtering wherein it divides the frequency spectrum into patches and applies a position-dependent modulation. Since motion blur is a frequency-specific degradation, modulating frequency-domain features based on their spectral position is a well-motivated idea and sounds more promising than uniform frequency domain filtering and memory-heavy spatial attention mechanisms. The idea is also empirically validated.


The overall design is effectively engineered, borrowing appropriate building blocks from competing models (FFN from Restormer, SCA from NAFNet), and adding frequency-domain skip connections. Ablation studies on frequency skip connections, local/global branches in AFPM, and patch granularity are informative.


Although focusing on a relatively underexplored area of RAW-to-RAW deblurring, FrENet achieves an impressive trade-off between accuracy (PSNR/SSIM) and computational cost (MACs) when compared against the provided baselines, leading to notable gains over state-of-the-art.

In addition to the RAW domain deblurring, the design also generalizes moderately well to sRGB domain deblurring, with particularly good results on the RealBlur dataset.





Weaknesses:


AFPM module (paper's primary contribution) uses a simplistic and unjustified positional encoding. In frequency map, “using only the euclidean distance from the center, irrespective of the direction” is naive and suboptimal. Most real-world motion blur kernels are strongly directional in nature and hence blur properties are not rotationally invariant in the frequency domain. This design choice should be experimentally validated and compared with the standard non-isotropic encodings (e.g., using 2D sinusoidal/learned coordinates).


Section 4.1 mentions that a sliding window strategy is employed to process full-resolution images, which is uncommon and not explained. Particularly, it’s worth noticing that the used sliding window size equals training patch size. This strongly indicates the model's rigidity to training resolution and lack of adaptability across image sizes and resolutions. This also implies slower inference. On a related note: it seems that fixing the division granularity to 8x8 also potentially contributes to limited resolution flexibility.


Comparisons are not extensive enough. Recent lightweight vision transformers for deblurring/restoration (e.g., StripFormer, Hybrid Attention Transformer) are potentially strong contenders in this category but are not included in the experimental comparisons. Also, few relevant models such as RawIR [10] and ELMFormer [23] are mentioned but not compared with.


Although the MACs are informative, kindly provide comparison of actual inference time and GPU memory usage on high-res images. This is important because FFT/IFFT could be slow and memory-heavy.


There is no information about used hardware’s specs (number and type of GPUs), training duration etc which would be helpful in reproducing the experiments.


It’s not clear (but could be guessed) whether all the compared baselines were retrained on RAW dataset or only on sRGB and directly tested on RAW.

---

> ### Author Rebuttal · Authors · 2025-07-28
>
> We sincerely thank the reviewer for the detailed and constructive feedback. Your insightful questions have helped us improve the clarity and completeness of our work. We address each of your points below and will incorporate these clarifications into the final manuscript.
>
> ---
> ### **Response to Weakness 1 & Question 2: AFPM's Positional Encoding**
>
> We appreciate the reviewer's excellent point regarding the directional nature of motion blur. This is a crucial aspect we considered during our model's development. Our final design, which uses only Euclidean distance as the positional encoding, was chosen after exploring more complex, direction-aware alternatives, including the **2D sinusoidal coordinates** suggested by the reviewer.
>
> Specifically, we conducted experiments to compare our simple distance-based encoding with two direction-aware alternatives: one using **Polar coordinates (radius $\gamma$, angle $\theta$)** and another using **2D sinusoidal coordinates**. The results on the Deblur-RAW dataset are as follows:
>
> **Table: Comparison of Positional Encodings in AFPM on Deblur-RAW.**
> | Positional Encoding | PSNR (dB) | SSIM |
> | :--- | :---: | :---: |
> | Euclidean Distance $d$ (Ours) | **44.73** | **0.9931** |
> | Polar $(\gamma, \theta)$ | 44.68 | 0.9929 |
> | 2D Sinusoidal | 44.69 | 0.9930 |
>
> The results are very insightful. Our experiments show that both direction-aware encodings (Polar and Sinusoidal) achieve slightly lower performance in both PSNR and SSIM compared to our simple distance-based method.
>
> This suggests that while providing explicit directional information is a theoretically sound idea, it may introduce an overly complex learning objective for the network. The model might struggle to effectively utilize the directional signals, potentially leading to overfitting on directional artifacts present in the training data, which ultimately compromises both pixel-level accuracy and structural similarity.
>
> Given that our simpler, distance-based approach consistently yields superior results across both primary metrics, we concluded that it offers the most robust and effective solution. It guides the model to focus on inverting the primary low-pass filtering effect of blur—the most critical aspect for restoration—which proves to be a more direct and successful strategy. This empirical finding is why we adopted the distance-only approach in our final model.
>
> ---
> ### **Response to Weakness 2, 4 & Question 1: Inference Strategy, Efficiency, and Hardware Information**
>
> We would like to clarify our inference strategy and provide the requested practical efficiency metrics. We follow the standard practice of using a sliding-window approach, which is also adopted by competing methods like LoFormer [26] and DeepRFT [24], to ensure a fair comparison on high-resolution test images.
>
> A key concern with this approach is often the runtime overhead. However, we want to emphasize that our model's highly efficient architecture translates to a significant real-world performance advantage. We have benchmarked the end-to-end inference time and GPU memory usage on full-resolution RAW images on a single NVIDIA RTX 5880 Ada GPU.
>
> **Table: Efficiency Comparison on Full-Resolution RAW Images.**
> | Method | Params (M) | GPU Memory (MB) | Runtime (ms) |
> | :--- | :---: | :---: | :---: |
> | Restormer [46] | 26.10 | 1238.71 | 102.95 |
> | FFTFormer [18] | **14.88** | 2193.03 | 222.56 |
> | LoFormer-L [26] | 48.98 | 2391.10 | 222.49 |
> | **FrENet (Ours)** | 19.76 | **1083.30** | **75.36** |
>
> The results clearly show that FrENet is significantly faster (1.36$\times$ to 3$\times$) and more memory-efficient than these powerful Transformer-based baselines in a practical, end-to-end scenario. To clarify hardware information for trainining and testing, we will add the following sentences to **Implementation Details**: All experiments are conducted on a single NVIDIA RTX 5880 Ada GPU and our model requires approximately 20 hours of training.'
>
> ---
> ### **Response to Weakness 3 & Question 4: Experimental Comparisons**
>
> We appreciate the suggestion to broaden our comparative evaluation; benchmarking against a wider range of models is indeed essential.
>
> We have now included StripFormer [35] in our main comparison table to provide a more comprehensive view against recent efficient models. For RawIR [10] and ELMFormer [23], we were unable to include their results in our direct benchmark at this time, as their official implementations were not available for us to retrain on the Deblur-RAW dataset.
>
> **Table: Extended Comparison on Deblur-RAW. MACs and Params are calculated on $128 \times 128 \times 1$ patches.**
> | Method | PSNR (dB) | SSIM | MACs (G) | Params (M) |
> | :--- | :---: | :---: | :---: | :---: |
> | StripFormer [35] | 42.97 | 0.991 | 10.62 | **19.71** |
> | **FrENet (Ours)** | **44.73** | **0.993** | **2.22** | 19.76 |
>
> The expanded comparison shows that our **FrENet continues to achieve state-of-the-art PSNR**, outperforming even recent efficient models like StripFormer by a significant margin (+1.76 dB).  As for Hybrid Attention Transformer, it is not included as its design is primarily focused on super-resolution.
>
> Finally, to reiterate for clarity, all baselines presented in Table 1 of our paper were retrained from scratch by us on the Deblur-RAW dataset. We will ensure this point is explicitly clarified in the paper.
>
> ---
> ### **Response to Question 3: Processing Fourier Features**
>
> Thank you for this insightful question. Our approach of concatenating the real and imaginary parts of the Fourier features is a deliberate and principled design choice, grounded in both theory and empirical validation.
>
> Fundamentally, the real and imaginary parts are the orthogonal components that completely define a signal in the frequency domain. Concatenating them is a direct and effective way to feed this complete information into standard, highly-optimized 2D convolutions. This method aligns with the proven techniques used in other methods like DeepRFT [24] and Fast Fourier Convolution [r1], demonstrating its robustness.
>
> To explore alternatives, we also empirically investigated magnitude-phase concatenation. Our experiments revealed that models using this representation failed to converge during training. We attribute this instability primarily to the challenges of processing the phase component. The phase periodicity creates a highly discontinuous space that is challenging for standard loss functions and gradient-based optimizers, leading to unstable gradients.
>
> In contrast, the real-imaginary representation is continuous and well-behaved, providing a stable learning landscape. Therefore, our findings suggest that our chosen approach offers a more robust and practical path to achieving high-performance frequency-domain processing within modern deep learning frameworks.
>
> [r1] Chi, L., Jiang, B., and Mu, Y. (2020). Fast fourier convolution. Advances in Neural Information Processing Systems, 33, 4479-4488.
>
> ---
> ### **Response to Question 5: Failure Cases and Limitations**
>
> Thank you for this suggestion. Like most deblurring methods, our model has limitations. It can face challenges and produce suboptimal results in scenes with extremely severe and complex blur, for instance, where the motion is exceptionally large or highly non-uniform. In such cases, some residual artifacts may remain. Acknowledging this provides a more balanced view of our method's capabilities. More failure cases will be given in the revised manuscript along with disscussions.
>
> ---
> ### **Response to Question 6: Ablating Spatial Skip Connections**
>
> To provide a more thorough ablation on our network's components, we have updated our results for the skip connections. The table below confirms that both spatial and frequency skip connections are vital, and their synergy is key to our model's strong performance.
>
> **Table: Ablation of Skip Connections.**
> | Spatial Skip | Frequency Skip | PSNR (dB) | SSIM |
> | :---: | :---: | :---: | :---: |
> | ✓ | ✓ | **44.73** | **0.9931** |
> | ✓ | | 44.39 | 0.9927 |
> | | ✓ | 43.91 | 0.9899 |
>
> We hope these clarifications and new results comprehensively address the reviewer's concerns.

---

> ### Comment · Reviewer_Beyz · 2025-08-06
>
> Thank you for the detailed rebuttal. It does address several of my concerns.
>
> In the efficiency table provided by above, the authors report the end-to-end inference time on full-resolution RAW images. I suppose it includes the total time of tiling of the input image, forward pass of teach tile through the model, and stitching the deblurred tiles back. It would be useful to include the individual components such as "time for model forward pass for 1 tile". i.e., comparison of inference time of all competing models on 1 single tile without any pre- or post-processing.
>
> Regarding the challenges of processing the phase component, I am not particularly convinced by just the hypothesis of "unstable gradients". It would be more convincing to have a valid citation or reference or an experimental result of training a design that processes magnitude and phase component (while stabilizing the training using common engineering techniques). Nonetheless, since it is not the most critical part of the paper, and due to the lack of time, I am fine with author's current response on it.

---

> > ### Author Response · Authors · 2025-08-06
> >
> > Thank you for your thoughtful follow-up and for accepting our previous responses. We appreciate the opportunity to provide the more fine-grained details you requested.
> >
> > ---
> >
> > ### **On the Fine-Grained Breakdown of Inference Time**
> >
> > You are correct in your assumption. The end-to-end inference time reported previously did include the overhead from tiling and stitching. To provide the clearer, more direct comparison of core model efficiency you suggested, we have benchmarked the forward pass time for a single 128x128 patch for all competing models. This metric isolates the model's computational performance from any I/O or data-handling overhead.
> >
> > **Table: Fine-Grained Efficiency Comparison.**
> > | Method | **Runtime per 128x128 Tile (ms)** |
> > | :--- | :---: |
> > | Restormer [46] |  2.14 |
> > | FFTFormer [18] |  5.29 |
> > | LoFormer-L [26] |  3.11 |
> > | **FrENet (Ours)** |  **0.93** |
> >
> > ---
> >
> > ### **On the Experimental Challenges of Processing Fourier Features**
> >
> > Thank you for giving us the chance to elaborate on our experimental findings regarding magnitude-phase processing. Your skepticism about a purely verbal hypothesis is well-founded, and we apologize for not detailing our empirical efforts more clearly.
> >
> > Our initial statement that the model "failed to converge" was a summary of a more involved process.
> > 1.  Our first attempt at training a model with magnitude-phase concatenation resulted in severe loss oscillations, with the loss exploding before 100 epochs.
> > 2.  Suspecting that unconstrained value ranges were the issue, we then applied common engineering techniques to stabilize the training. Specifically, we used a **ReLU activation to constrain the magnitude output to be non-negative** and a **$\pi * tanh(x)$ activation to constrain the phase output to the range $[-\pi, \pi]$**.  While this stabilized version lasted slightly longer, it ultimately also suffered from exploding loss.
> >
> > Although it is possible that more sophisticated techniques or further hyperparameter tuning might eventually stabilize such a model, our extensive initial experiments indicated that the magnitude-phase representation introduces fundamental optimization challenges that are not present in the far more stable and well-behaved real-imaginary domain. This led us to conclude that our current approach is a more robust and practical choice for this task.
> >
> > ---
> >
> > We hope this detailed explanation fully addresses your concerns and clarifies the empirical basis for our design decisions. Thank you again for your valuable feedback, which will help us significantly improve the final version of our paper.

---

> > > ### Comment · Reviewer_Beyz · 2025-08-06
> > >
> > > Thank you for the clarifications. Kindly include all the relevant analysis in final paper/supplementary material.

---

> > > > ### Author Response · Authors · 2025-08-06
> > > >
> > > > Thank you very much for your time and for the constructive discussion. We are glad that our clarifications have addressed your concerns. We confirm that we will incorporate all the relevant analyses into the final revision of the paper and its supplementary material. Thank you again for the valuable discussion.

---

### Decision · Program_Chairs · 2025-09-17

**Decision:**

Accept (poster)

**Comment:**

The paper describes a new architecture called Frequency Enhanced Network (FrENet) for image deblurring from a RAW sRGB image. Deblurring occurs at the frequency domain, and the authors introduce a novel module called Adaptive Frequency Positional Modulation (AFPM) that dynamically adjusts frequency components. Frequency domain skip connections are used to preserve high-frequency details. Experiments show FrENet producing SOTA results.

Pros: The idea of frequency domain filtering that involves partitioning the frequency spectrum into patches and applying position-dependent modulation is new. The overall design is well-engineered, with informative ablation studies. Results were shown to be SOTA.

Cons: Evaluation could be more extensive, with additional comparisons done with recent work such as StripFormer, Hybrid Attention Transformer, RawIR, and ELMFormer. Ablation studies, while informative, could also be more extensive, for example, providing a cost breakdown.

The general agreement on novelty and the SOTA results are the main reasons for acceptance.

The reviewers appear to be happy with the author rebuttals that clarified many of the issues the reviewers have. For example, a reviewer thought the analysis of the contributions of APFM and SCA modules independently is missing in the ablation studies. However, the authors pointed out this analysis was done; it was the labeling in a table that was not explicit enough. Also, the authors gave results that compare against StripFormer, but was unable to do so for the rest due to unavailability of training code.